# Dress&Dance:
## DRESS UP AND DANCE AS YOU LIKE IT

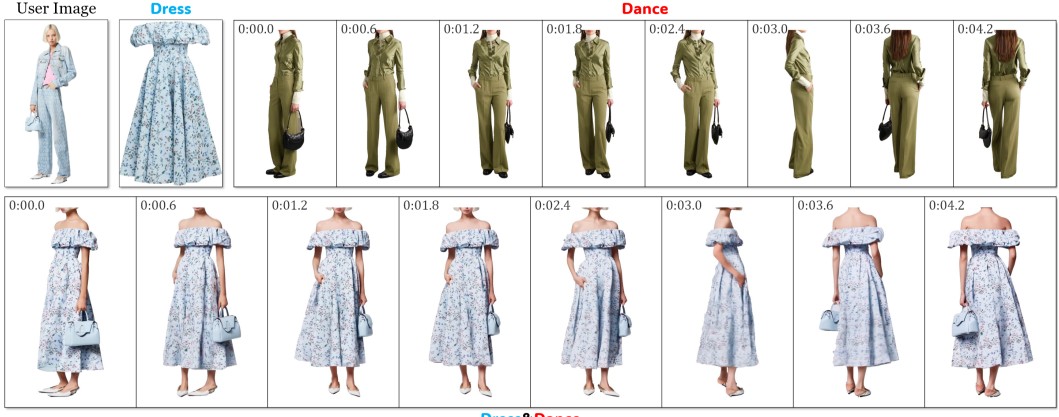

Figure 1: Given a single image of a user, a garment (*dress*) that they would like to wear, and an example video showing how they would like to animate themselves (*dance*) as shown in the first row. Our method, **Dress&Dance**, generates a high-quality $5s$ video ($1152 \times 720$, 24 FPS) of the user wearing the target garment containing desired motion while maintaining their accessories such as bag and shoes. Video results are provided in our Supplementary Video (**SV**, demo.mp4).

## ABSTRACT

We present **Dress&Dance**, a video diffusion framework that generates high-quality virtual try-on videos of users wearing desired garments while performing complex motions. Our approach is the *first* to achieve high-resolution ($1152 \times 720$) at high-FPS (24 FPS), with support for various try-on modes, including simultaneous try-on of tops and bottoms. At the core of our framework is *CondNets*, a novel attention-based conditioning architecture that unifies heterogeneous multi-modal inputs – including garment images, user images, motion videos, and text prompts – into a single homogeneous token sequence. To prevent strong pre-trained text priors from overshadowing garment inputs, we introduce *garment-aware target steering*, a guidance mechanism that enforces accurate garment placement. To further address both data scarcity and computational demands of training high-quality video models, we propose a *synthetic triplet generation* strategy for producing paired training data and a *multi-stage training curriculum* that progressively scales resolution and frame rate. Our framework outperforms existing open-source and commercial solutions, enabling flexible, high-quality try-on experiences that faithfully preserve garment details, user identity, and complex motions.

## 1    INTRODUCTION

Imagine trying on a new garment in a fitting room: it is natural to move around to see how the fabric drapes and flows. However, current computational methods for virtual try-on typically generate only static 2D images, failing to deliver an immersive or expressive experience. In this paper, we introduce Dress&Dance, a video diffusion framework that generates high-quality try-on videos of users wearing desired garments – 5-second clips at 24 FPS and $1152 \times 720$ resolution. Motion is

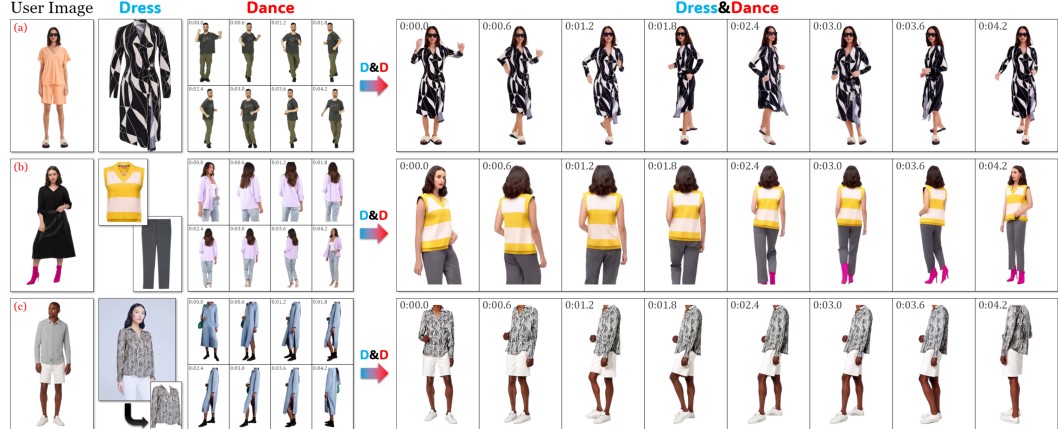

Figure 2: Dress&Dance allows for flexible virtual try-on, including (a) complicated dancing motions, (b) simultaneous try-on of tops and bottoms, and (c) transferring a garment from another user.

guided by a user-provided reference video, allowing the user to both choose the garment (*Dress*) and control how they animate with it (*Dance*), as shown in Fig. 1. Dress&Dance supports a wide range of tops, bottoms, and one-piece garments, as well as simultaneous try-ons of multiple garments. It further enables trying on clothes borrowed from another person or modifying garments with text prompts. Figure 2 illustrates these applications.

Our framework builds upon advances in diffusion models, which have enabled impressive text-to-video (Gu et al., 2023; Yang et al., 2024d) and image-to-video (Blattmann et al., 2023; Zhang et al., 2023b; Kling, 2024) generation, but remain fundamentally limited for video try-on. While text-to-video models offer some controllability, they struggle to preserve essential details and maintain the integrity of the target garment and user appearance without highly specific descriptions. An alternative strategy is to perform single-image try-on and then animate the resulting image with a video diffusion model. However, this often produces temporally incoherent outputs due to error propagation from the first frame, as shown in Fig. 3-(a). Another difficulty lies in generating nuanced motion: while simple movements can be described with text, complex motions are challenging to convey, even for state-of-the-art commercial models like Kling (2024) and Ray2 (2025), as seen in Fig. 3-(b). Our work addresses these issues with an end-to-end framework that leverages a reference video to guide intricate motion. As shown in Fig. 3-(c), the results of Dress&Dance faithfully preserve garment and user appearance details, even when parts of the garment are occluded in the original image, while accurately following the motion of the reference video.

A core technical obstacle in realizing such an end-to-end framework is the simultaneous handling of heterogeneous, multi-modal inputs, including garment images, user images, motion videos, and text prompts. Prior try-on approaches typically rely on separate conditioning strategies for each modality, resulting in brittle and non-scalable architectures. This challenge is compounded by the fact that, in real-world scenarios, these modalities are *rarely pixelwise-aligned* with the generated video – for example, garment images are often provided as flat product photos rather than aligned to the user image. As a result, widely adopted conditioning models such as ControlNet (Zhang et al., 2023a), which assume pre-aligned inputs, are ill-suited for our task. To address this, we propose a novel attention-based conditioning architecture, *CondNets*, which unifies all heterogeneous inputs into a single, *homogeneous* token sequence through attentive integration. Even with unified conditioning, however, another challenge emerges: strong pre-trained text priors dominate attention, causing the model to overlook garment inputs and fail at accurate garment placement. To counter this, we introduce *garment-aware target steering*, an *explicit guidance* mechanism that balances text dominance and enforces garment alignment while preserving motion fidelity.

While our architectural design addresses the challenges of multi-modal conditioning, scaling the framework to generate high-resolution, high-FPS try-on videos introduces further challenges in training. A key bottleneck is data: paired examples of the same person wearing different garments in identical poses are particularly scarce. Previous methods often circumvent this by relying on garment-agnostic masks or other intermediate representations, which are brittle, prone to artifacts, and create a mismatch between training and inference (detailed in Sec. 2). To overcome this, we propose a *synthetic triplet generation* strategy that leverages pre-trained models to produce high-quality

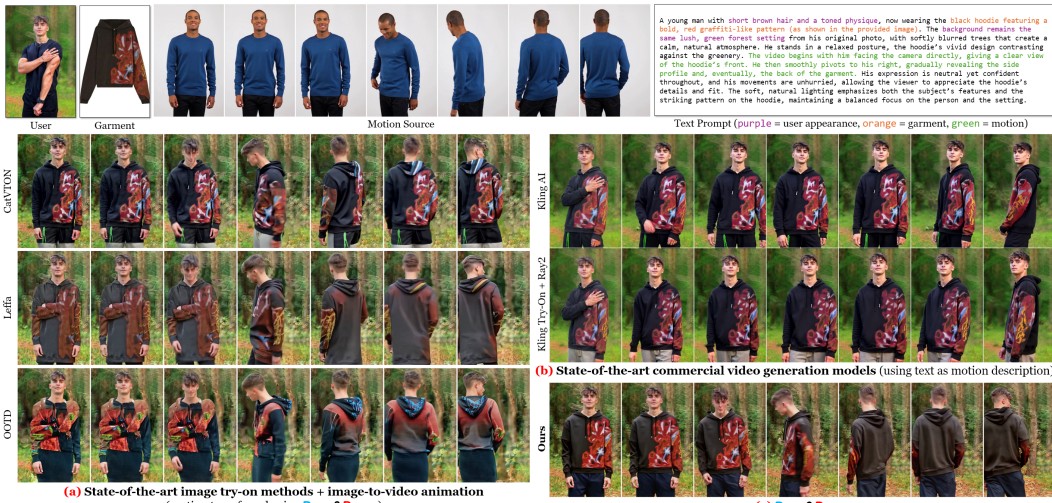

Figure 3: Given a user image, a desired garment, and a reference video, we extract a detailed text description via GPT (OpenAI, 2023) as shown in the first row. (a) State-of-the-art single image try-on methods struggle to generate correct try-on, whilst error propagation happens when motion is applied. (b) State-of-the-art commercial models, such as Kling, are able to perform try-on but still struggle in capturing the nuanced motion description, as *text* alone is not sufficient for motion description. Also, as the user's right hand covers part of the garment, the information of the covered patterns is lost for the video generation model, leading to incorrect garment appearances when the hand moves in the video, for all baselines in (a) and (b). (c) Our **Dress**&**Dance** generates high-quality virtual try-on results that faithfully preserve both garment and user appearance with precise motion, even when the hand moves off. Video results are provided at **SV**3:19.

paired training data. In addition, the computational demands of synthesizing long, high-resolution videos are prohibitive. To alleviate this, we adopt a *multi-stage training curriculum* that progressively increases spatial resolution and frame rate, enabling the model to scale-up efficiently.

**Our contributions** are threefold: (1) We propose *CondNets*, a novel conditioning framework that unifies heterogeneous multi-modal inputs through cross-attention, together with a guidance mechanism, *garment-aware target steering*, which ensures accurate garment fitting. (2) We introduce efficient training strategies, including *synthetic triplet generation* and a *multi-stage training curriculum*, to address data scarcity and computational cost, enabling high-resolution, high-FPS video model training with limited resources. (3) We demonstrate state-of-the-art performance in high-fidelity virtual try-on, robustly handling complex motions and diverse garment types, and outperforming both open-source and commercial baselines such as Kling (2024) and Ray2 (2025).

## 2 RELATED WORK

**Image-to-Video Generation** and **Video-to-Video Generation.** Please refer to Appendix Sec. C.

**Image-to-Image Virtual Try-On.** The dominant line of virtual try-on research focuses on generating a single image output from a user image and one or more garment references (Han et al., 2018; Wang et al., 2018; Issenhuth et al., 2020; Yang et al., 2020; Lewis et al., 2021; Ge et al., 2021a; Choi et al., 2021; Morelli et al., 2022; Bai et al., 2022; Dong et al., 2022; He et al., 2022; Lee et al., 2022b; Yang et al., 2022; Li et al., 2023b; Morelli et al., 2023; Xie et al., 2023; Yan et al., 2023; Zhu et al., 2023b; Li et al., 2023a; Yang et al., 2024b; Choi et al., 2024; Zhang et al., 2024a; Kim et al., 2024; Li et al., 2024; Xu et al., 2025; Chong et al., 2025; Wang et al., 2024a; Kim et al., 2024; Yang et al., 2024a; Xu et al., 2024a; Xiel et al., 2023; Lee et al., 2022a; Yang et al., 2024c; Zhu et al., 2023a; Ge et al., 2021b; Chen et al., 2023; Zhu et al., 2024a; Xing et al., 2025). Many adopt a two-step design: garment warping followed by synthesis using GANs (Goodfellow et al., 2014) or diffusion models (Rombach et al., 2022a; Podell et al., 2023; Peebles & Xie, 2023). While effective, the reliance on paired training data introduces information leakage, and intermediate garment-agnostic representations propagate artifacts. Some methods aim to jointly modify pose

and garment (Raj et al., 2018; Han et al., 2019; Sarkar et al., 2020; Cui et al., 2021; Sarkar et al., 2021), but they typically lack temporal smoothness when extended to sequences. Our approach sidesteps these limitations by synthesizing unpaired training triplets, aligning training and inference settings, and avoiding intermediate representations.

**Video Virtual Try-On (VVT).** Moving beyond single images, video-based try-on aims to generate temporally coherent sequences conditioned on garments. Early approaches (Dong et al., 2019; Zhong et al., 2021; Jiang et al., 2022; Kuppa et al., 2020) relied on GANs and frame warping to approximate motion. With the rise of diffusion models, subsequent works (Nguyen et al., 2025; Xu et al., 2024b; Wang et al., 2024b; Fang et al., 2024; He et al., 2024b; Karras et al., 2024) explored both image-to-video and video-to-video pipelines. A straightforward strategy is to first synthesize a try-on image and then animate it using an image-to-video module (Chan et al., 2019; Siarohin et al., 2019; Holynski et al., 2021; Siarohin et al., 2021; Xu et al., 2024c; Guo et al., 2024; Karras et al., 2023; Zhang et al., 2024b; Azadi et al., 2023; Tevet et al., 2022; Guo et al., 2022; Hu, 2024; Zhu et al., 2024b). However, such modular pipelines suffer from error accumulation across stages. More integrated methods employ end-to-end video generation to preserve garment details while enforcing temporal consistency (He et al., 2024a).

While these methods have advanced the field, they often face significant challenges in practice. First, they struggle to effectively handle the heterogeneous nature of multi-modal inputs. Existing approaches for image or video try-on frequently design disparate conditioning strategies for each modality, leading to complex and non-scalable architectures. For instance, some use channel concatenation for pose maps while relying on attention for garments. Furthermore, general conditioning models like ControlNet (Zhang et al., 2023a) are ill-suited for diffusion transformer (DiT) (Peebles & Xie, 2022) architectures and pixelwise-unaligned inputs such as a flat garment image, which requires complex, non-local transformations to be placed on the final video (more details in Appendix Sec. D). Second, due to the scarcity of paired training data (the same person wearing different garments in the same pose), conventional methods rely on multiple intermediate representations like garment-agnostic masks to convert the ground truth into a valid input. This process can lead to information loss, artifacts from imprecise masks, and a mismatch between the training and inference settings. Furthermore, although recent models like Fashion-VDM (Karras et al., 2024) and Swift-Try (Nguyen et al., 2025) produce higher resolution outputs, they remain restricted to short clips (around 5s) and typically below $720 \times 540$ resolution. In contrast, our Dress&Dance scales video virtual try-on to long, high-resolution sequences ($1152 \times 720$, 121 frames) by leveraging a unified conditioning mechanism, a robust training strategy that avoids intermediate representations, and a DiT-based video diffusion architecture with joint motion and garment control.

## 3 Dress&Dance: Methodology

High-fidelity video virtual try-on requires synthesizing a dynamic video from a complex set of heterogeneous, non-aligned inputs: one or more flat garment images $g$, a static user image $I_{u,g'}$ of user $u$ wearing some other garments $g'$, a motion reference video $m$, and an optional text prompt $p$. The model is expected to generate $V_{u,g}$, a video of the user $u$ wearing the desired garment $g$, while performing the indicated motion from $m$ and elsewhere matching the description in $p$. The core challenge lies in how to effectively and robustly integrate these distinct modalities and faithfully reflect them in the final high-resolution video.

We introduce Dress&Dance, a novel video diffusion framework that explicitly tackles this challenge by framing it as two fundamental problems: (1) how to unify diverse multi-modal conditions, and (2) how to ensure the model properly attends to all the inputs. We address these with our novel and universal *CondNets* architecture. Furthermore, we design specialized training methods to efficiently utilize limited video data, and employ a multi-stage curriculum learning strategy to progressively and rapidly achieve high-resolution, high-FPS generation.

### 3.1 CondNets: A Unified and Attentive Conditioning Framework

Generating a high-fidelity virtual try-on video from multiple, distinct condition inputs necessitates a conditioning mechanism that can gracefully and effectively integrate them. To address this, we propose a novel conditioning network, *CondNets*, which leverages the inherent flexibility of the

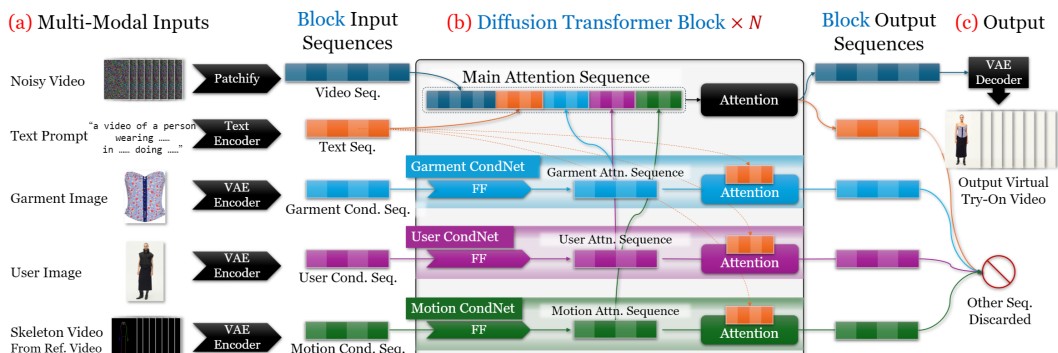

Figure 4: **Dress&Dance** accounts for multi-modal conditioning through our unified CondNets architecture based on the attention mechanism. The "Block Input Sequences" and "Block Output Sequences" represent the sequences passed recursively from one block to the next. At the final stage, only the video sequence is decoded by the VAE to generate the output virtual try-on video, while other sequences are discarded.

attention mechanism to unify all heterogeneous inputs into a single, homogeneous sequence. This approach provides a straightforward, general, and extensible solution for multi-modal conditioning, overcoming the architectural complexities and limitations of previous methods.

**Homogenizing Heterogeneous Inputs.** Our framework is built upon the Diffusion Transformer (DiT) architecture (Peebles & Xie, 2022), which natively treats video and text as sequences of tokens and fuses them via attention. Our key insight is to extend this mechanism by converting every conditional input into a dedicated attention sequence that can be seamlessly incorporated into the DiT's denoising process. As visualized in Figs. 4 and G.1, CondNets is implemented as a parameter-shared copy of the DiT's layers, augmented with LoRA adaptors (Hu et al., 2022) for each condition. This network processes each conditional input—such as a user image, a garment image, or a motion skeleton video—and generates a corresponding attention sequence at each layer. These conditional sequences are then concatenated with the video and text sequences within each DiT block for attention fusion. This design allows the model to implicitly connect every pixel of the conditions to the output video frames through cross-attention, which is crucial for transferring fine-grained garment and user details. This unified mechanism even enables condition-specific optimizations. For motion guidance, we preprocess the reference video to extract a sparse skeleton sequence, which significantly reduces computational overhead by focusing only on the human body parts.

**Dominance of Text-Priors vs. Garment Conditioning.** Attention-based conditioning mechanisms, including our *CondNets*, face a common training challenge where the model's powerful pre-trained text-guided generation prior tends to dominate. This is manifested in the attention computation, where pre-existing text prompt tokens often receive dominantly high attention scores, causing them to overshadow other conditions. Consequently, the model struggles to properly incorporate newly added conditions. This issue is particularly problematic for the garment condition, which necessitates fitting the garment image onto the user across various poses, demanding the model not only to correctly place the garment but also to infer and generate unseen views (e.g., its back or side). During conventional training, the model often learns to minimize the loss by directly overfitting to the text prompt and the reference video, rather than acquiring the ability to properly handle the garment. Consequently, the model struggles with the challenging garment-to-body try-on task, or more specifically, the "garment placement." This phenomenon is a general and even more severe problem for larger-scale diffusion models with stronger pre-trained priors.

**Garment-Aware Target Steering Guidance.** To address this severe attention dominance problem and enable the model to learn the complex garment fitting task, we design a three-phase *garment-aware target steering* to guide the model. This targeted strategy presents simplified video generation tasks in each phase, gradually forcing the model to rely on the garment image. Our method utilizes specialized *synthetic videos* as temporary training targets (different from the ground truth try-on video) to guide the model's focus. The core manipulation is a content isolation technique where we significantly reduce the contrast of irrelevant video content via a linear combination with a pure color (e.g., $0.1 \cdot$ original $+ 0.9 \cdot$ gray), effectively "de-emphasizing" or "graying out" those areas, as shown in Fig. 5. This manipulation creates a clearer learning signal for the model. In the first phase, we construct the target video by placing the flat garment image as an overlay *in front of*

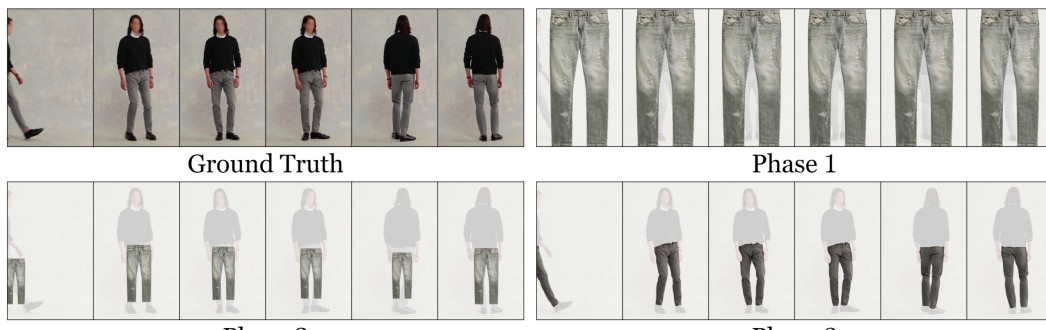

Figure 5: Our three-phase *Garment-Aware Target Steering Guidance* is designed to mitigate the attention dominance problem by enforcing the model to attend to the garment condition. This guidance progressively steers the model's training target from a simplified try-on task to a complex one, ensuring it correctly learns the correspondence between the video content and the garment image to develop robust try-on capability.

each ground truth frame, while de-emphasizing the non-occluded surrounding background. This approach forces the model to prioritize copying the raw content from the large, static garment image. In the second phase, we advance this target by positioning the flat garment as an overlay at the garment's correct occurrence location and size within each ground truth frame. This encourages the model to learn the spatial association between the garment and the human body. In the third phase, the target video uses the garment segmented from the ground truth frame, with all other parts de-emphasized, teaching the model to learn the full and complex garment placement on the body. This strategy allows the model to learn to generate correct try-on videos based on the garment after a fast-converging training of approximately one to two hours per phase, a capability that cannot be achieved by a 200-hour conventional training without this guidance. Crucially, while this guidance focuses on the garment, we observe that the resulting model also shows significant improvements in handling other conditions, such as the user image, demonstrating the effectiveness and broad benefit of this tailored steering strategy.

## 3.2 TRAINING STRATEGY

While our *CondNets* architecture and curriculum learning strategy provide a robust solution for multi-modal conditioning of virtual try-on, two key challenges remain on the training side. First, high-quality paired video data is extremely limited. Second, generating high-resolution, high-FPS videos requires a significant computational budget, which makes training costly and time-consuming. To address these issues, we introduce a set of synergistic training strategies that maximize data utility and enable efficient training.

**Synthetic Triplet Generation.** Supervised training of our model requires triplets $(g, I_{u,g'}, V_{u,g})$ of a garment image(s) $g$, a user image $I_{u,g'}$ wearing a *different* garment $g'$, and a target video $V_{u,g}$ of the same user wearing garment $g$. However, our datasets consist primarily of paired data, such as $(g, V_{u,g})$, which lacks the crucial unpaired user image $I_{u,g'}$ needed for training. To circumvent this, we create *synthetic* triplets by leveraging various pre-trained image try-on models. We perform an image try-on on a random frame from $V_{u,g}$ using a randomly selected garment $g'$ to directly generate the required unpaired user image $I_{u,g'}$. This strategy ensures a close alignment between our training and inference settings, eliminating the need for error-prone intermediate representations.

**Scaling up Conditions, Resolution, and FPS with Curriculum Learning.** Generating high-resolution, high-FPS videos with diffusion models from multiple conditions requires synthesizing a massive number of tokens, which is computationally intensive and time-consuming. To accelerate convergence and reduce training costs, we adopt a synergistic multi-stage training strategy, framed as *curriculum learning*, instead of training at full resolution with all conditions from the start. We progressively scale the task complexity along two axes: *conditions* and *resolution/FPS*. We first build upon a text-to-video base model, gradually adding complexity by introducing motion, garment, and user image conditions in subsequent stages. Concurrently, we progressively scale the video resolution while maintaining the full $1152 \times 720$ resolution for image conditions. This design differentiates our approach from standard progressive scaling methods like Gu et al. (2024) and Kar-

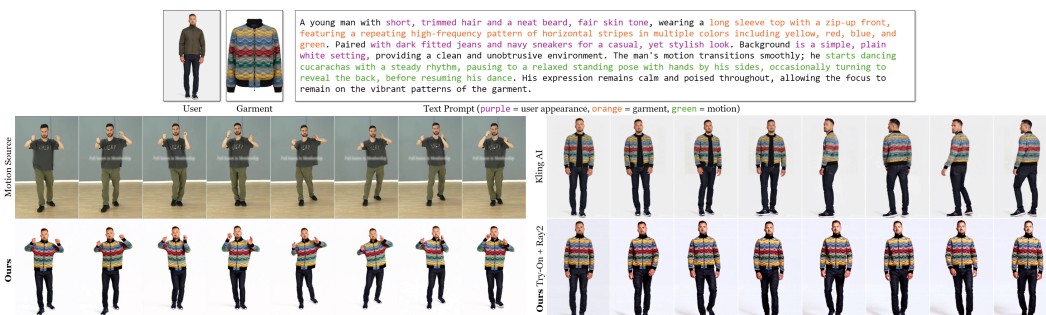

Figure 6: Dress&Dance allows a user to **dress** themselves in a desired garment and perform the desired **dance**. These complex movements are challenging to express with text alone, which makes motion generation difficult for models like Kling (Kling, 2024) and Ray2 (Ray2, 2025).

| Method Property | | Method | Evaluation with Ground Truth on Captured Dataset | | | |
|---|---|---|---|---|---|---|
| Ours | Commercial | | PSNR↑ | SSIM↑ | LPIPS$_{VGG}$ ↓ | LPIPS$_{AlexNet}$ ↓ |
| | | TPD (Yang et al., 2024a)  + CogVideoX I2V | 14.47 | 0.8305 | 0.2840 | 0.2461 |
| | | OOTD (Xu et al., 2024a)  + CogVideoX I2V | 14.68 | 0.8282 | 0.2779 | 0.2457 |
| | | ML-VTON  + CogVideoX I2V | 14.49 | 0.8270 | 0.2961 | 0.2520 |
| ✓ | | **Dress&Dance** Image Try-On + CogVideoX I2V | 17.26 | 0.8515 | 0.2812 | 0.1635 |
| | ✓ | Kling Image Try-On  + Kling Video 1.6 | 17.33 | 0.8651 | **0.2296** | 0.1683 |
| ✓ | ✓ | **Dress&Dance** Image Try-On + Kling Video 1.6 | 15.21 | 0.8297 | 0.2835 | 0.2066 |
| ✓ | ✓ | **Dress&Dance** Image Try-On + Ray2 | 15.16 | 0.8290 | 0.2938 | 0.1990 |
| ✓ | | **Dress&Dance**, Direct Train | 17.14 | 0.8678 | 0.2854 | 0.1338 |
| ✓ | | **Dress&Dance** | **22.41** | **0.9038** | 0.2382 | **0.0624** |

Table 1: In quantitative evaluations, Dress&Dance significantly outperforms open-source baselines on most metrics while achieving comparable or superior quality metrics to commercial models like Kling (2024) and Ray2 (2025).

ras et al. (2024), which are typically used to learn generative capabilities from scratch. Unlike those methods, simply scaling down all inputs would disrupt the pre-trained feature space of our foundation model. Our video-image hybrid strategy allows us to inject control signals efficiently without suffering from catastrophic forgetting of the model's robust spatial and temporal priors. This allows the model to quickly learn the core task at lower resolutions (e.g., $768 \times 480$) without compromising its ability to handle high-resolution inputs or 41-frame videos in later stages. The full training regimen is detailed in Appendix Sec. G.5.

Additionally, to address the limited 8 FPS of our base model, we introduce a subsequent training stage, fine-tuning the model from its final-stage resolution output. In this stage, the model is trained to upsample the videos from 8 FPS to a smooth 24 FPS while significantly enhancing visual quality. We achieve this by intentionally applying various data augmentations during training to simulate potential defects such as blurriness and color distortion. This strategy equips the model with a powerful ability to correct and refine the final output, effectively serving as the final step in our progressive training regimen. More details about this stage are in Appendix Sec. H.

**Video-Image Hybrid Training.** Image data is much easier to obtain than video data – as a comparison, the number of garments in our video dataset is only 2% of that of our image dataset. To leverage this data and benefit our video generation training, we adopt a hybrid strategy in which each batch consists of both video and image data. Our *CondNets* architecture natively supports this, making the model extensible for these heterogeneous batches. In the early low-resolution training stages, we also utilize full $1152 \times 720$ resolution image data to maintain the model's high-resolution capability. This approach incurs minimal additional cost compared to video-only training while allowing Dress&Dance to benefit from the abundant image data throughout the training process.

**Fast Inference.** To improve generation efficiency, we propose two methods to reduce inference time without compromising quality. We introduce a classifier-free guidance (CFG, Rombach et al. (2022b)) distillation training, adapting the approach described in Meng et al. (2023), and fine-tune our model to support one-step denoising. With these optimizations, our Dress&Dance can generate high-quality videos in less than 10 minutes, significantly accelerating the user experience. The full details are in Appendix Sec. I.

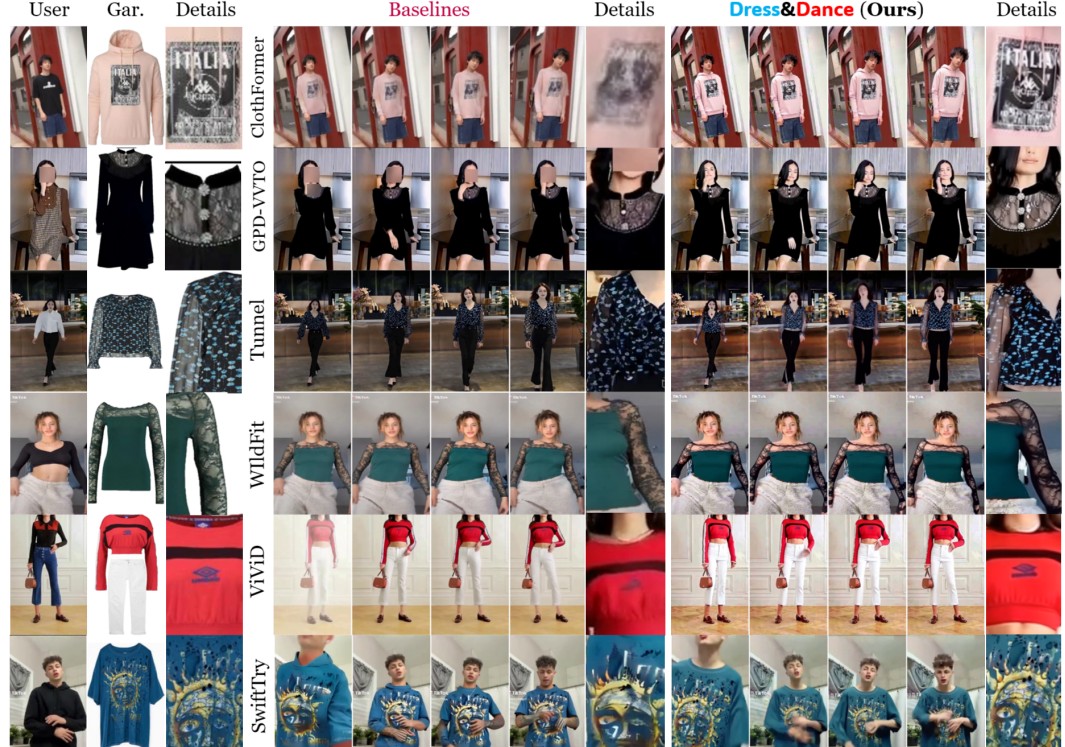

Figure 7: Dress&Dance significantly outperforms existing video virtual try-on methods, with much more detailed and precise textures and better support for transparent garments.

## 4 EXPERIMENTS

We evaluate the performance of Dress&Dance across three different try-on modes (Fig. 2): (1) the *single garment mode* allows a user to try on a garment from a single flat image; (2) the *multiple garment mode* supports the simultaneous try-on of all provided flat garment images; and (3) the *garment transfer mode* enables transferring a garment from an existing image via segmentation.

**Datasets.** Since existing video virtual try-on datasets (Dong et al., 2019; Fang et al., 2024; Nguyen et al., 2025) are limited to lower resolutions ($\leq 720 \times 540$), we curate two high-resolution ($\geq 1080$P) video datasets for both training and evaluation. (1) The *Internet video dataset* is constructed by crawling publicly available fashion videos paired with flat garment images, resulting in approximately 80K garment-video pairs. (2) The *captured video dataset* is created by hiring 183 human models to record try-on videos for various sets of garments. Each model records videos for around 100 different garment sets, which allows us to construct unpaired multi-garment try-on data with ground truth by cross-matching frames of different garments worn by the same person. We also collect an *image dataset* from the Internet with around 4M image pairs for hybrid training. We divide each dataset into training and test subsets and perform training on the combination of all training subsets. For evaluation, we sample garments and models from the evaluation subsets or utilize other garments, user images, and motion reference videos sourced from the Internet.

**Baselines.** To provide a comprehensive comparison, we evaluate our approach against two main categories of baselines. We tailor evaluation focus, *e.g.* quantitative or qualitative, based on each baseline's code availability.

(1) *Video virtual try-on (VVT) baselines.* We first compare Dress&Dance against VVT methods, ViViD (Fang et al., 2024), WildFit (He et al., 2024b), Tunnel Try-On (Xu et al., 2024b), GPD-VVTO (Wang et al., 2024b), and ClothFormer (Jiang et al., 2022) in the single-garment setting. For the aligned garment transfer task, we compare with to Fashion-VDM (Karras et al., 2024). For a fair comparison, and due to the lack of publicly available code for some baselines, we compare with their results presented on their respective project websites.

(2) *Image try-on + Image-to-video generation.* We also compose baselines using a two-stage framework, which involves applying a state-of-the-art image try-on method to the user's image and then animating the result with a separate video generation model. For image try-on, we use state-of-the-art open-source and commercial methods, including Kling Try-On (Kling, 2024), CatVTON (Chong et al., 2025), Leffa (Zhou et al., 2024), TPD (Yang et al., 2024a), OOTD (Xu et al., 2024a), and ML-VTON (a combination of GP-VTON (Xiel et al., 2023) and HR-VTON (Lee et al., 2022a)), along with our own Dress&Dance to showcase its single-frame try-on capability. The animated videos are then generated by top open-source models like CogVideoX 1.5 I2V (Yang et al., 2024d) for text-guided animation, or our Dress&Dance for motion-guided animation. We did not use other methods like Stable Video Diffusion (Blattmann et al., 2023) or I2VGen-XL (Zhang et al., 2023b) as they do not support portrait videos well. We also apply this strategy with two state-of-the-art commercial models for animation: Kling Video 1.6 (Kling, 2024) and Ray2 (Ray2, 2025).

**Qualitative Comparison.** Our qualitative results are showcased in Figs. 3, 6, K.1, and K.2. Figure 3 demonstrates the results in the single garment try-on mode, with more results available in our **SV**. A key failure case for baseline methods is evident in Fig. 3: the user's hand occludes the top-right corner of the garment in the input image. Any baseline that relies on an "image try-on followed by animation" pipeline, even if animated by powerful commercial models Kling (2024) or Ray2 (2025) or our Dress&Dance, fails to reproduce the correct garment pattern once the hand moves. Since the animation model lacks access to the original garment image throughout the process, the occluded information is permanently lost and cannot be recovered. In stark contrast, Dress&Dance processes the garment image throughout the entire video generation, enabling it to produce correct, temporally consistent, and high-fidelity video try-on results. Furthermore, our model excels in handling complex, non-textual motion guidance. In Fig. 6 and our **SV**, we evaluate challenging dancing motions that are difficult to describe with text alone, leading to the failure of all text-conditioned baselines. By leveraging a sparse skeleton video extracted from the reference dance, our Dress&Dance accurately generates smooth, intricate motions alongside correct try-on synthesis.

The performance in the multiple garment mode is illustrated in Fig. K.2. Dress&Dance is capable of simultaneously performing virtual try-on for both a top and a bottom garment *without* requiring any explicit labeling of garment type (*e.g.*, top or bottom). Regardless of the input order or garment types, Dress&Dance consistently generates high-quality, compositionally accurate try-on results. Conversely, the Kling AI baseline, despite officially supporting multi-garment try-on, incorrectly tries on the trousers as a skirt in this challenging scenario.

We compare Dress&Dance with multiple video try-on baselines in single garment mode in Fig. 7, and in the garment transfer mode in Fig. K.1. Dress&Dance consistently generates videos at a significantly higher resolution ($1152 \times 720$), which is crucial for preserving the fine-grained details and textures of the garments. Our method also exhibits superior quality in rendering challenging semi-transparent garments. In comparison, the results from all other baselines are typically limited to a lower resolution (*e.g.*, $720 \times 540$), often resulting in blurred and subdued textures.

**Quantitative Comparison.** Tab. 1 presents our quantitative comparison on tasks constructed from the captured dataset. We compare the generated videos with the ground truth using PSNR, SSIM, and LPIPS (Zhang et al., 2018). Dress&Dance outperforms most methods and remains highly competitive with commercial models, including Kling Video 1.6 (Kling, 2024) and Ray2 (Ray2, 2025).

For the Internet dataset, evaluating try-on and visual quality is challenging due to the high degrees of freedom in generation. Inspired by VQAScore (Lin et al., 2024), we leverage the strong vision-language reasoning capabilities of GPT (OpenAI, 2023) to grade the generated videos. We focus on the following aspects for evaluation: garment try-on fidelity ($GPT_{Try-On}$), user appearance fidelity ($GPT_{User}$), human and garment motion quality ($GPT_{Motion}$), visual quality ($GPT_{Visual}$), and an overall quality score ($GPT_{Overall}$). To guide GPT's grading, we provide detailed instructions and rubrics as Appendix Figs. K.3 and K.4. Each video is graded 40 times, and the average score is taken to reduce randomness and ensure fairness. The results are shown in Table E.1. Dress&Dance demonstrates superior virtual try-on capability, achieving the highest garment fidelity ($GPT_{Try-On}$) scores across all models. In terms of overall visual quality, our approach remains highly competitive with commercial baselines despite their significant data advantage and simpler motion constraints. The results confirm Dress&Dance's ability to maintain high fidelity of both the garment and user identity during complex, detailed motion synthesis. The detailed analysis is in Appendix Sec. E.

**Ablation Study.** We conduct an ablation study to analyze the effectiveness of our training strategy, including the *Garment-Aware Target Steering Guidance* and *Curriculum Learning*. We define a "Direct Training (DT)" variant for Dress&Dance, which trains directly on the final-stage inputs and outputs at full resolution, without incorporating the specialized strategies detailed in Secs. 3.1 and 3.2. We compare the full Dress&Dance with the DT variant in Table 1 and in our **SV**. Without our targeted *Garment-Aware Target Steering Guidance* and progressive *Curriculum Learning*, the model fails to faithfully preserve the content of both the user and garment images, even after extended training. This results in poor quantitative metrics. Our results demonstrate that our specialized training strategy is crucial for the model's convergence and final performance.

## 5 CONCLUSION

In this work, we presented Dress&Dance, a novel video diffusion framework that achieves the first high-resolution ($1152{\times}720$), high-FPS (24 FPS) video virtual try-on with complex motion guidance. Central to our contribution is CondNets, which serves as the foundational architectural backbone enabling the effective fusion and alignment of heterogeneous modalities from text to images to video – a structural capability that existing work cannot achieve. Complementing this architectural core, we introduced critical methodological innovations: the three-phase garment-aware target steering guidance to resolve learning difficulties such as attention dominance, and a synergistic multi-stage training curriculum to overcome data scarcity and prevent catastrophic forgetting. Collectively, this holistic system sets a new state of the art for the field, demonstrating superior garment fidelity and motion consistency significantly beyond existing solutions.

## ETHICS STATEMENT

The technology presented in this paper, Dress&Dance, has the potential for significant positive societal impact, primarily by enhancing consumer confidence in e-commerce. By providing highly dynamic and accurate virtual try-on, our model is expected to substantially decrease the rate of product returns in online clothing retail, promoting both business efficiency and environmental sustainability by reducing shipping and waste.

However, as a powerful generative tool for human video, this framework is inherently susceptible to misuse. The capability to synthesize realistic videos of individuals wearing arbitrary clothing could be exploited to create misleading content, generate deepfakes, or produce outputs that perpetuate bias or harmful stereotypes if the underlying training data is not perfectly balanced. We affirm that the purpose of this work is strictly academic research. We strongly advocate that any future deployment of models based on this work must include robust safeguards, clear watermarking protocols, and strict ethical guidelines to actively prevent deceptive use and ensure responsible, socially conscious application of the technology.

Regarding the construction of our captured dataset, we adhered to strict ethical standards for all 183 participants involved. First, regarding **conditions**, data collection was conducted in a fully equipped professional studio environment. To ensure the safety and comfort of the participants, each subject was assisted by a dedicated support team of five professionals, including photographers and stylists. Second, we obtained explicit **informed consent** from all participants; each subject signed a formal service agreement prior to the session, authorizing the data capture and its usage for research purposes. Finally, regarding **compensation**, all subjects were fairly and generously compensated for their time and effort. The compensation rates were highly competitive (comparable to the hourly rates of senior engineering roles) to ensure the ethical treatment of all individuals involved.

## REPRODUCIBILITY STATEMENT

We are committed to ensuring the reproducibility of our work and facilitating future research. To this end, we will provide the following upon acceptance: (1) The benchmark utilized in this paper, along with all raw video outputs generated by Dress&Dance, enabling direct and fair comparison for subsequent work. (2) Code to allow the community to build upon our methods. (3) Detailed documentation of hyperparameters, training schedules, and implementation details, as Appendix Secs. G,F,H, and I to ensure faithful reproduction of our experiments.

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

# Dress&Dance: Appendix

## A    Supplementary Video (SV)

We provide a supplementary video (SV) as `demo.mp4` to visualize our videos and comparisons against baselines. Here is a table of contents of the SV:

- **SV**0:00: Introduction and try-on modes.
- **SV**0:15: Single garment try-on.
- **SV**2:02: Multiple garment try-on.
- **SV**2:48: Garment Transfer.
- **SV**3:09: Text-only try-on.
- **SV**3:14: Comparison with baselines and ablation study.
- **SV**3:29: FPS Upsampling and Artifact Removal (Appendix Sec. H).
- **SV**3:34: Fast inference (Appendix Sec. I).

In the following sections, we use **SV** to refer to this supplementary video.

## B    ICLR: LLM Use Statement

We utilize large language models (LLMs), *e.g.*, GPT (OpenAI, 2023), in the following ways:

**Paper Writing Refinement.** We use LLMs to refine our paper writing and correct grammar mistakes.

**Evaluation Metrics.** We use LLMs to implement some of our evaluation metrics in Tab. E.1.

## C    Related Work: Video Generation

**Image-to-Video Generation.** A representative method, Stable Video Diffusion (SVD) (Blattmann et al., 2023), generates short videos from a single image but is limited to landscape formats and short durations. Similarly, I2VGen-XL (Zhang et al., 2023b) takes an image and text prompt to produce short landscape videos. A more flexible solution, CogVideoX-I2V (Yang et al., 2024d), supports portrait videos and integrates text guidance for controllable synthesis. Commercial systems, such as Kling Video 1.6 (Kling, 2024) and Ray2 (Ray2, 2025), also enable portrait image-to-video generation with text input, offering customizable outputs. These methods highlight progress in animating still images but do not address garment try-on or motion-driven control. We note that Kling also provides a separate image try-on model, which we analyze in this work.

**Video-to-Video Generation.** A related body of work investigates video editing and translation for general content manipulation. For example, VideoShop (Fan et al., 2024) propagates edits from the first frame to subsequent ones using a diffusion model, and BIVDiff (Shi et al., 2024) refines frame edits to improve temporal consistency. CogVideoX-V2V (Yang et al., 2024d) combines a pre-trained text-to-video diffusion model with SDEdit (Meng et al., 2022) for text-driven video translation. While these methods demonstrate the power of temporal editing, they do not address virtual try-on specifically. Our approach unifies garment replacement and motion-guided generation, closing this gap.

## D    CondNet versus ControlNet (Zhang et al., 2023a)

Our CondNets shares a conceptual similarity with ControlNet (Zhang et al., 2023a) in that both strategies reuse parameters from the main diffusion model. However, the mechanisms of CondNet and ControlNet are fundamentally distinct. ControlNet injects conditions into the main generation process by *additively* fusing feature maps from its network branch with the main feature map on

| Method Property | | Method | | Evaluation of Try-On and Visual Quality | | | | | | |
|---|---|---|---|---|---|---|---|---|---|---|
| Ours | Commercial | | | $GPT_{Try-On}$ ↑ | $GPT_{User}$ ↑ | $GPT_{Motion}$ ↑ | $GPT_{Visual}$ ↑ | $GPT_{Overall}$ ↑ | $FID_{Internet}$ ↓ | $FID_{Captured}$ ↓ |
| | | TPD (Yang et al., 2024a) | + CogVideoX I2V | 69.67 | 73.98 | 68.15 | 65.45 | 68.64 | 1146 | 753 |
| | | OOTD (Xu et al., 2024a) | + CogVideoX I2V | 70.57 | 76.41 | 68.71 | 70.78 | 70.76 | 1089 | 884 |
| | | ML-VTON | + CogVideoX I2V | 69.69 | 76.62 | 68.50 | 69.95 | 70.50 | 1185 | 739 |
| ✓ | | **Dress&Dance** Image Try-On + CogVideoX I2V | | 85.05 | 87.54 | 75.68 | 76.88 | 80.71 | 1109 | 760 |
| | ✓ | Kling Image Try-On | + Kling Video 1.6 | 80.10 | **89.97** | 85.48 | *84.70* | 84.38 | **982** | **655** |
| ✓ | ✓ | **Dress&Dance** Image Try-On + Kling Video 1.6 | | *86.85* | 89.77 | **82.59** | **84.94** | **85.85** | *1008* | 700 |
| ✓ | ✓ | **Dress&Dance** Image Try-On + Ray2 | | 86.79 | 88.99 | 79.31 | 83.48 | 84.18 | 1094 | 735 |
| ✓ | | **Dress&Dance**, Direct Train | | 79.48 | 78.47 | 72.00 | 71.24 | 74.85 | 1073 | 788 |
| ✓ | | **Dress&Dance** | | **87.41** | 88.89 | *80.35* | 84.48 | *84.95* | 1055 | *691* |

Table E.1: Our Dress&Dance significantly outperforms all baselines in garment fidelity in virtual try-on ($GPT_{Try-On}$), demonstrating superior try-on capability. It also achieves highly comparable or even better visual quality on other metrics to powerful commercial baselines like Kling Video 1.6 (Kling, 2024) and Ray2 (Ray2, 2025), which outperform all open-source baselines by a large margin.

| Benchmark | Method | $GPT_{Try-On}$ ↑ | $GPT_{User}$ ↑ | $GPT_{Motion}$ ↑ | $GPT_{Visual}$ ↑ | $GPT_{Overall}$ ↑ |
|---|---|---|---|---|---|---|
| ClothFormer | ClothFormer | 65.0 | 72.4 | 75.2 | 60.6 | 68.8 |
| | **Dress&Dance** | **90.6** | **88.5** | **85.0** | **87.4** | **88.5** |
| GPD-VVTO | GPD-VVTO | 70.9 | 75.8 | 78.4 | 65.1 | 72.1 |
| | **Dress&Dance** | **92.4** | **90.9** | **87.1** | **88.1** | **89.2** |
| Tunnel Try-On | Tunnel Try-On | 74.2 | 86.0 | **82.0** | 69.2 | 74.8 |
| | **Dress&Dance** | **76.9** | **88.6** | 80.0 | **75.4** | **81.6** |
| ViViD | ViViD | 72.2 | 78.9 | 78.1 | 68.6 | 84.0 |
| | **Dress&Dance** | **85.6** | **87.3** | **86.3** | **84.6** | **85.2** |
| WildVidFit | WildVidFit | 82.4 | 83.4 | 77.1 | **75.8** | 81.0 |
| | **Dress&Dance** | **86.0** | **86.4** | **80.2** | 74.6 | **81.4** |
| SwiftTry | Swifttry | 70.8 | 70.9 | 72.7 | 56.0 | 65.2 |
| | **Dress&Dance** | **83.5** | **83.1** | **78.6** | **75.2** | **80.3** |

Table E.2: Quantitative comparison with VVT baselines. Our Dress&Dance consistently outperforms existing methods across the majority of metrics.

a pixel-by-pixel basis. This approach is well-suited for *pixel-aligned* conditions, where each pixel in the conditional input corresponds to the same location in the generated output (e.g., depth maps, Canny edges, and contour lines). However, this additive fusion method fundamentally struggles with *pixel-unaligned* conditions, such as a flat garment image, where the spatial relationship between the conditional image and the target video requires complex, non-local transformations.

In contrast, our CondNets injects conditions by concatenating the condition sequences with the corresponding main sequences. By leveraging the attention mechanism, CondNets is able to deal with *pixel-unaligned* conditioning far more effectively through the cross-attention computed between all pairs of tokens across the entire concatenated sequence.

Additionally, ControlNet was specifically designed for UNet-based diffusion models, relying on their U-shaped skip-connection architecture. It typically models only the downsampling part of the UNet, requiring roughly half the parameters and computation of a full network. For a DiT-based diffusion model, a corresponding ControlNet would need to be a full, multi-layer transformer structure, significantly increasing the computational overhead and making it unsuitable for our scalable framework.

# E  QUANTITATIVE RESULT ANALYSIS

The quantitative results are shown in Tables E.1, E.2, and E.3.

## E.1  OUR BENCHMARK: VIRTUAL TRY-ON CAPABILITY.

As shown in the "$GPT_{Try-On}$" row in Table E.1, our Dress&Dance significantly outperforms all baseline models in garment fidelity and try-on quality. Furthermore, using our Dress&Dance for the image try-on step in the baselines' methods consistently improves the try-on quality by a large margin compared to other image try-on methods, including Kling AI (Kling, 2024). This demonstrates our model's superior capability in virtual try-on.

| Method | User$_{\text{Try-On}}$ ↑ | User$_{\text{User}}$ ↑ | User$_{\text{Motion}}$ ↑ | User$_{\text{Visual}}$ ↑ | User$_{\text{Overall}}$ ↑ |
|---|---|---|---|---|---|
| Prefer **Dress&Dance** to Kling | 77.0 | 77.3 | 86.5 | 76.5 | 77.5 |
| Prefer **Dress&Dance** to Ray2 | 77.0 | 76.5 | 85.5 | 75.0 | 77.0 |
| Prefer **Dress&Dance** to Fashion-VDM | 84.8 | 78.8 | - | 78.8 | 78.8 |
| Prefer **Dress&Dance** to ClothFormer | 85.0 | 73.5 | - | 78.0 | 78.0 |
| Prefer **Dress&Dance** to GPD-VVTO | 77.0 | 76.5 | - | 70.0 | 70.0 |
| Prefer **Dress&Dance** to Tunnel | 74.3 | 63.8 | - | 63.0 | 63.0 |
| Prefer **Dress&Dance** to WildVidFit | 73.3 | 67.3 | - | 66.5 | 67.0 |
| Prefer **Dress&Dance** to ViViD | 76.3 | 67.5 | - | 69.5 | 68.8 |
| Prefer **Dress&Dance** to SwiftTry | 61.3 | 70.5 | - | 60.5 | 60.5 |

Table E.3: User study preference rates comparing our Dress&Dance against various baselines. Each value represents the weighted preference rate of Dress&Dance over the specific baseline; values $> 50\%$ indicate a preference for our result. Note that the User$_{\text{Motion}}$ comparison is excluded for standard VVT baselines, as they do not support independent motion reference inputs.

### E.2 OUR BENCHMARK: VISUAL QUALITY.

In all other metrics of Table E.1, our Dress&Dance achieves results highly comparable to the commercial models. It is important to note that commercial models train with significantly more video data and have the flexibility to generate motion only constrained by a text prompt. This makes it intrinsically easier for them to achieve high scores with fewer constraints on motions, but they are unable to perfectly reproduce the intricate motion from a reference video. Even so, our Dress&Dance still achieves very similar scores on all metrics. It is worth highlighting that the top-performing baseline in our evaluation, which achieves the best results in some metrics, is a combination of our image try-on capability with Kling's animation

### E.3 VIDEO VIRTUAL TRY-ON (VVT) BASELINES

In Table E.2, we compare our Dress&Dance with VVT baselines using benchmarking tasks directly sourced from their respective official websites to ensure a fair comparison. The results demonstrate that our Dress&Dance outperforms each baseline in most aspects, achieving consistently superior try-on quality, user appearance consistency, and overall video quality.

However, it is important to note that low-resolution benchmarks inherently favor low-resolution baselines. The reduced resolution and blurriness tend to mask potential visual and motion artifacts, and standard metrics often lack explicit penalties for low resolution. Consequently, some baselines may achieve artificially higher scores in motion and visual smoothness simply because their imperfections are concealed by the low-fidelity output.

### E.4 USER STUDY

In Table E.3, we report the results of a user study comparing our Dress&Dance against each baseline. For each comparison, participants were presented with the input user image, garment image, and motion reference, and asked to indicate their preference based on five criteria: User$_{\text{Try-On}}$ (faithfulness to the garment), User$_{\text{User}}$ (preservation of user identity), User$_{\text{Motion}}$ (fidelity to the reference motion), User$_{\text{Visual}}$ (overall visual quality), and User$_{\text{Overall}}$ (comprehensive assessment). Note that for the User$_{\text{Motion}}$ metric – which assesses how faithfully the generated video adheres to the reference motion – we restricted the evaluation to comparisons with general generative baselines (Kling and Ray2). Standard VVT baselines are excluded from this evaluation because they do not support independent motion reference inputs. Since these methods rely on processing the original source video frames rather than synthesizing motion from a driving signal, this specific motion-generation metric is inapplicable to their workflow. We adopted a weighted scoring system where users selected a degree of preference from {Strongly Prefer, Prefer, Slightly Prefer}, corresponding to weights of 5, 3, and 1, respectively. The final preference rate is calculated as the ratio of the weighted sum favoring our method to the total weighted sum.

Figure F.1: Our Dress&Dance supports text-only try-on. By introducing a different garment and accessories in the text prompt, our Dress&Dance generates try-on results without corresponding images.

We conducted this user study with 40 participants. As shown in the table, our method achieves a preference rate exceeding 50% against all baselines, demonstrating that Dress&Dance is consistently preferred by users.

## F PROMPT GENERATION AND TEXT-ONLY TRY-ON

### F.1 OBTAINING TEXT PROMPTS

Our Dress&Dance requires a text prompt to provide semantic guidance for the video generation. When other image and video conditions are present, this text prompt can be easily generated using a vision-language model (VLM) like GPT (OpenAI, 2023). We follow a simple multi-step process: first, we send sampled frames of the reference video to the VLM and request a detailed description of the motion. Then, we send the user and garment images, asking the VLM to generate a detailed prompt of the form: "the person in the user image, changing the garment to the one in the garment image, performing the motion as described in the motion description." This composite approach generates a plausible and comprehensive text prompt. In practice, since most of the critical information is contained in the image and video conditions, our Dress&Dance remains highly effective even with a very simple text description.

### F.2 TEXT-ONLY TRY-ON

In our Dress&Dance, the text prompt serves a crucial function as a fallback or refinement mechanism: if certain required information is not fully described in other conditions, the details provided in the text will be utilized. This capability allows our framework to achieve text-only try-on without any garment images. As shown in Figure F.1, our Dress&Dance supports text-guided virtual try-on, enabling users to specify a garment and even add new accessories purely through the prompt.

## G IMPLEMENTATION DETAILS

### G.1 DETAILED ARCHITECTURE

Figure G.1 visualizes the detailed architecture of our diffusion transformer integrated with Cond-Nets for multi-modal conditioning. The CondNet for a specific condition, e.g., the garment image, contains LoRA-augmented (Hu et al., 2022) parameter-shared copies of each transformer block. At every transformer block, the main sequence is constructed by concatenating the video sequence, the text sequence, and the condition sequences (specifically, the key and value sequences from the attention). The model then computes only the output video and text sequences. The output condition sequences for each transformer block are computed in their respective CondNet branches through cross-attention with the text sequence.

### G.2 BASIC DESIGNS

Our main video diffusion model generates a 41-frame, 8-FPS video at $1152 \times 720$ (resulting in a video length of approximately 5 seconds, consistent with prior work like CogVideoX (Yang et al., 2024d) which generates a 49-frame, 8-FPS video of 6 seconds). The subsequent upsampling model

Figure G.1: Our Dress&Dance designs. (a) Dress&Dance supports multi-modal conditioning through our CondNets architecture, a unified conditioning framework augmented with LoRA (Hu et al., 2022)-parameter-shared transformers. (b) The two core models in our Dress&Dance pipeline: a main model for 8-FPS video generation, and an upsampling model to auto-regressively increase the frame rate to 24-FPS while removing artifacts.

generates a 25-frame, 24-FPS video at $1152 \times 720$. All image conditions are maintained at a resolution of $1152 \times 720$. The skeleton videos are ViTPose (Xu et al., 2022) outputs at the same resolution and length as the target video, featuring a gray background and a customized skeleton coloring to clearly distinguish left and right arms and legs.

## G.3 CONDITIONING NETWORKS

We utilize LoRA (Hu et al., 2022) adaptors to enable the heterogeneous behavior required for different conditions. Specifically, we apply LoRA adaptors not only in the "to query/key/value" linear layers of the attention modules but also within the feed-forward layers and convolutional layers. The rank of LoRA is set to 64 to ensure sufficient capacity.

## G.4 POSE CONDITION IMPLEMENTATION

To provide precise motion guidance and resolve geometric ambiguities (e.g., distinguishing frontal from dorsal views), we employ a modified skeleton representation based on ViTPose (Xu et al., 2022). We introduce three specific enhancements to the standard visualization scheme to enforce structural consistency:

**Auxiliary Midpoints and Central Axis.** We introduce three additional auxiliary keypoints representing the midpoints of the eyes, shoulders, and hips. We connect these midpoints to their respective left and right anatomical keypoints (e.g., connecting the left shoulder to the shoulder midpoint) using edges with distinct colors. This explicitly defines the body's central axis and symmetry plane.

**Lateral Color Differentiation.** We implement a strict color separation strategy for the body topology. The edges representing the left limbs (arms, legs) and the left side of the torso are rendered in contrasting colors to their counterparts on the right side. This provides the model with unambiguous strong signals regarding the subject's orientation.

**Robust Hand Representation.** Unlike standard visualizations that may assign unique colors to individual fingers, we unify the color of all finger edges on a specific hand to match the color of the corresponding arm (e.g., all left fingers share the left arm's color). This simplification reduces the visual complexity of the condition map and improves the model's robustness against high-frequency jitter or errors in finger pose estimation.

**Compressing Skeleton Videos Into Sparse Sequences.** A naive approach for converting a skeleton video into a sequence is to treat it as a standard $F \cdot H \cdot W$ sequence, similar to ordinary videos. However, this dramatically increases computational costs—doubling the main sequence length and quadrupling the self-attention computation. We observe that skeleton videos contain numerous "empty" pixels (background regions without any part of the skeleton), accounting for roughly $80\%$ of each frame. Since our primary concern is the human pose, we convert the skeleton video into a sparse sequence of variable length by masking and preserving only the sequence elements that correspond to non-background regions. With positional embeddings tracking the location of each sparse element, our Dress&Dance is able to learn the correct motion from this sparse sequence (in contrast to a "dense" sequence containing all $F \cdot H \cdot W$ elements). This strategy significantly reduces computational costs by $70\%$ without compromising quality or conditioning effectiveness.

## G.5 TRAINING SETTING

We build our Dress&Dance architecture based on CogVideoX (Yang et al., 2024d)-5b. We reuse the parameters of the pre-trained CogVideoX model and initialize the new parameters related to Cond-Nets. We set separate learning rates for non-attention parameters ($4 \times 10^{-5}$), non-LoRA attention parameters ($10^{-5}$), and LoRA parameters ($2.5 \times 10^{-6}$). We use an AdamW optimizer with default hyperparameters and no scheduler.

We train our Dress&Dance on a single machine with 8 NVIDIA H100 GPUs. During the **Garment-Aware Target Steering Guidance**, we train Dress&Dance's main video diffusion model for 500 iterations for each of the three phases. Following this, we execute the **Curriculum Learning** in stages: we train the main video diffusion for 24,000 iterations in stage 1 at $768 \times 480$ resolution, and then 11,000 iterations in stage 2; afterward, we add the skeleton video condition and CondNet, and continue to train the main video diffusion for 6,000 iterations in stage 3 at $768 \times 480$ resolution, and then 6,000 iterations in stage 4 at full resolution to obtain our initial main video diffusion model. Then, initialized from this main video diffusion, we train our subsequent upsampling model for 12,000 iterations at full resolution, and continue to train the main diffusion model for another 12,000 iterations in stage 4.

## G.6 EVALUATION SETTING

For the Captured dataset with ground truth, we randomly sample 32 virtual try-on tasks from the test subset as our evaluation benchmarks. For the Internet dataset, we sample tasks by randomly pairing user images and garment images from the test subset, and constructed 32 single garment try-on tasks, 32 multiple garment try-on tasks, and 16 garment transfer tasks. We compute the Fréchet Inception Distance (FID) of a generated video with respect to a randomly sampled subset of 8192 videos from the corresponding dataset.

## H FPS UPSAMPLING AND ARTIFACT REMOVAL

Due to the capability limitations of the underlying CogVideoX base model, our Dress&Dance can only generate 41-frame videos at 8 FPS in one forward pass. To enable the generation of smoother videos at higher FPS, we introduce a subsequent upsampling model as the final stage in our **Curriculum Learning** (Sec. 3.2) to achieve 24 FPS. This upsampling model not only (1) converts the 41-frame videos into smooth 121-frame videos at 24 FPS but also (2) refines visual quality by removing defects such as contrast issues, color distortion, blurriness, and unnatural motions present in the videos generated by the base model. The upsampling model operates in an auto-regressive manner, refining the video segment by segment with overlaps. To train the upsampling model, we construct training data by temporally downsampling raw 24-FPS ground truth videos to 8 FPS and applying various data augmentations to the downsampled videos to simulate potential artifacts. This specialized training equips the upsampling model with the ability to remove artifacts and further enhance the final visual quality.

### H.1 AUTO-REGRESSIVE VIDEO GENERATION.

Due to limited GPU memory, we cannot directly generate a 121-frame video in a single forward pass during both training and inference. Therefore, we train the upsampling model to generate a 25-frame refined 24-FPS video from a 9-frame, 8-FPS original video, which we term the *source video* for refinement. The CondNets for this source video condition is designed by repeating each frame of the 8-FPS video three times, converting it into a non-smooth 24-FPS sequence, and then a standard $F \cdot H \cdot W$ sequence. To generate a 121-frame video, we divide it into seven segments of size 25, with an overlap of 9 frames between each pair of adjacent segments. We refine these segments progressively, leveraging DiffEdit (Couairon et al., 2022) to ensure consistent refinement across overlapped frames.

### H.2 DATA AUGMENTATION FOR VISUAL REFINEMENT TRAINING.

The upsampling model is trained within a standard diffusion training pipeline, where we construct 8-FPS video inputs by temporally downsampling the ground truth videos and train the model to restore the original video. We observe that, beyond temporal upsampling, the model also has the potential to refine visual quality by removing artifacts. Therefore, we retain all input conditions used to generate the original video, and then apply extensive data augmentations to the 8-FPS video inputs during training. These augmentations include color jitter, Gaussian blur, VAE compression (encoding and decoding the video with VAE, which may lose details), noising, local pixel shuffling (Zhong et al., 2023), and temporal blurring, to simulate various artifacts that may occur in the generated source video. With this robust training strategy, our upsampling model significantly enhances the visual quality of the final video, especially in addressing blurriness, contrast, and brightness issues.

### H.3 IMPLEMENTATION DETAILS

The full refined video is a 121-frame, 24-FPS video, refined from a 41-frame, 8-FPS video. We first repeat each frame three times and remove the last two frames to convert the 41-frame video into a coarse 121-frame version. Then, we refine this coarse video segment by segment. In the first step, we take frames $[0, 25)$ as the input to the upsampling model to generate the refined segment. In the subsequent steps, given the last refined index $i$ where frames $[0, i)$ are already refined, we take frames $[i-9, i+16)$ as the input to the upsampling model, where frames $[i-9, i)$ are preserved, and frames $[i, i+16)$ are newly refined. We apply DiffEdit (Couairon et al., 2022) on the first 9 frames to ensure they preserve the previously refined frames $[i-9, i)$, and the model generates smooth frames $[i, i+16)$ as the continuation. The entire refinement process is completed in 7 steps through 7 segments: $[0, 25)$, $[16, 41)$, $[32, 57)$, $[48, 73)$, $[64, 89)$, $[80, 105)$, and $[96, 121)$.

To grant the upsampling model the capability to re-generate blurred parts, it also takes all the conditions used by the main video diffusion model. More specifically, we directly pass the same garment image(s) and user image, and the corresponding segment of the skeleton video. The refinement source frames are also input to the upsampling diffusion model through an individual CondNet. This design ensures that the model can have heterogeneous behaviors to either preserve the first 9 input frames or refine the whole segment, without highly relying on DiffEdit to maintain smoothness.

## I EFFICIENCY OPTIMIZATION AND FAST INFERENCE

Our standard video generation strategy takes 100 denoising steps to generate a video in both main video generation and upsampling stages, resulting in a total generation time of roughly 4 hours for a 24-FPS video. We propose several techniques to optimize our generation, enabling us to produce videos in 3-5 minutes with similar quality.

### I.1 UNIPC MULTI-STEP SCHEDULER

Our standard generation process uses the DDIM (Song et al., 2020) scheduler, which relies only on the first-order derivative and requires numerous denoising steps. We replace this with the UniPC (Zhao et al., 2023) multi-step scheduler, which leverages higher-order derivatives to perform denoising based on the predicted noise in multiple steps. With UniPC, we are able to reduce the denoising

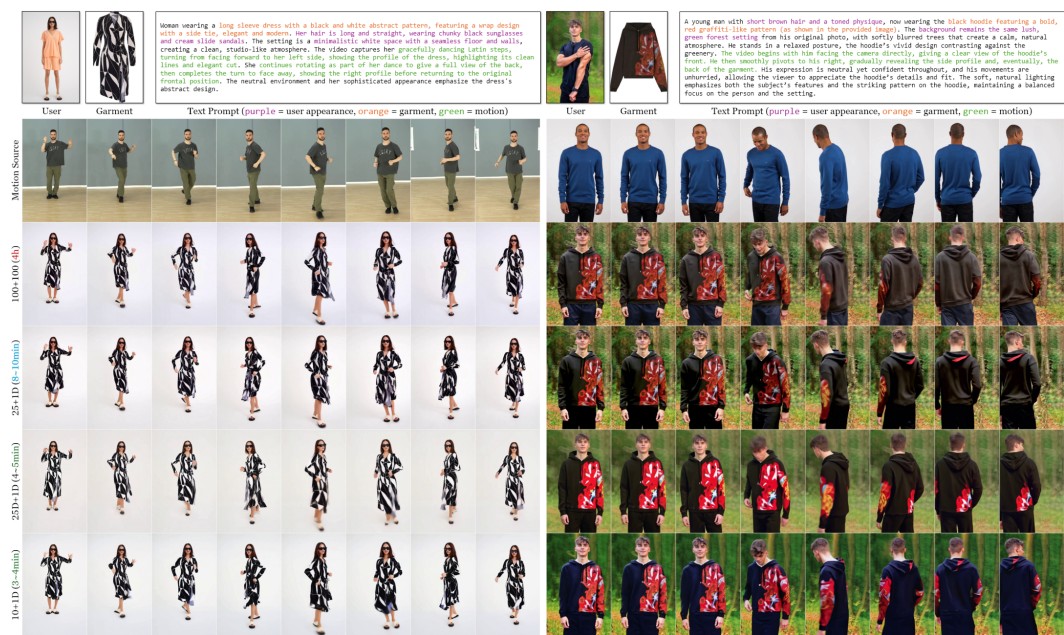

Figure I.1: Our Dress&Dance generates videos at highly-comparable quality in "25+1D" mode within only 8-10 minutes, while the "25D+1D" and "10+1D" modes generate slightly worse results with half generation time. More results are available in our **SV**.

steps from 100 to 10 for the main model and from 100 to 2 for the upsampling model while maintaining comparable quality. Notably, this replacement can be applied to a pre-trained model in an out-of-the-box manner without requiring specific training with the UniPC scheduler.

## I.2 CFG Strategy and Distillation

**CFG Strategy.** To handle multiple heterogeneous conditions effectively, we categorize them based on their guidance strength and conditioning difficulty into two sets:

- **Easy Conditions** ($\mathcal{C}_{\text{easy}}$)**:** Conditions that provide strong, unambiguous signals which the model follows easily. In our framework, this includes *motion*, *user image*, and the *source video* (in the refiner module).
- **Hard Conditions** ($\mathcal{C}_{\text{hard}}$)**:** Conditions requiring complex semantic interpretation or mapping, specifically *text prompts* and *garment images*.

We apply guidance scaling specifically to the hard conditions while treating the easy conditions as the unconditioned baseline. Let $f(\mathbf{z}_t, t, \mathcal{C})$ denote our DiT-based noise predictor network, where $\mathbf{z}_t$ is the noisy video latents at timestep $t$. We compute the joint noise prediction $\epsilon_{\text{all}}$ using all conditions $\mathcal{C}_{\text{all}} = \mathcal{C}_{\text{easy}} \cup \mathcal{C}_{\text{hard}}$, and the baseline prediction $\epsilon_{\text{easy}}$ using only easy conditions:

$$\epsilon_{\text{all}} = f(\mathbf{z}_t, t, \mathcal{C}_{\text{all}}), \quad \epsilon_{\text{easy}} = f(\mathbf{z}_t, t, \mathcal{C}_{\text{easy}}). \tag{1}$$

The final classifier-free guided noise $\epsilon_{\text{cfg}}$ is computed as:

$$\epsilon_{\text{cfg}} = w \cdot (\epsilon_{\text{all}} - \epsilon_{\text{easy}}) + \epsilon_{\text{easy}}, \tag{2}$$

where $w$ is the guidance scale (set to $w = 5.0$ in our experiments).

**Distillation.** Standard CFG necessitates two network evaluations per denoising step, doubling the inference latency. To eliminate this overhead, we employ a distillation approach building upon the framework introduced by Meng et al. (2023). We train a distilled model $g$ to directly predict the

guided noise $\epsilon_{\mathrm{cfg}}$ derived from our strategy above. The distillation objective is to minimize:

$$\mathcal{L}_{\mathrm{distill}} = \mathbb{E}_{\mathbf{z}_t, t, \mathcal{C}} \left[ \| g(\mathbf{z}_t, t, \mathcal{C}_{\mathrm{all}}) - \epsilon_{\mathrm{cfg}} \|^2 \right]. \tag{3}$$

By applying this distillation training to both the main video diffusion and the upsampling model, we enable the model to output the guided result in a single forward pass instead of two passes for $\epsilon_{\mathrm{all}}$ and $\epsilon_{\mathrm{easy}}$, effectively reducing the inference time by half.

### I.3 UPSAMPLING MODEL 1-STEP GENERATION

As the upsampling model's task is relatively simple and can already support 2-step generation in a training-free manner, we further train it for one-step generation. This training involves training the distilled model to output a noise prediction such that using it to denoise the noisy video obtains the target original video. Through this training, we achieve 1-step generation for the upsampling model.

### I.4 MODES AND RESULTS

Finally, we support three different modes to trade off between efficiency and quality. Each mode is marked as "$a\{,\mathrm{D}\} + b\{,\mathrm{D}\}$", e.g., "25D+1D" or "10+1D", representing a mode with $a$ main video denoising steps and $b$ upsampling model denoising steps, with ('D') or without (no 'D') CFG distillation. The three modes are as follows: (1) "25+1D" achieves videos at very similar visual quality in only 8∼10 minutes; (2) "25D+1D" and (3) "10+1D" modes achieve videos at only slightly lower quality in 4∼5 minutes, where the former offers better support for complicated garments and high-frequency detail generation, and the latter offers better user and garment fidelity. The results are visualized in Figure I.1 and in our **SV**.

## J LIMITATIONS

A limitation of our model is that its performance may degrade if the motion source video contains very fast or overly complicated motions. For example, if the motion source video is a very fast dance or artistic gymnastics, the model may fail to faithfully preserve the motion in the generated video, because (1) the main 8-FPS video might be too intermittent to accurately characterize the rapid motion, and (2) such motion might be unseen in our training dataset, preventing our model from accurately depicting it. One potential workaround is to fine-tune our model on the motion source video for several iterations to inject its prior into the model, allowing it to better mimic the source video's motion during generation. However, this process is time-consuming and may still not perfectly generate the video with the given motion.

## K ADDITIONAL FIGURES AND TABLES.

Due to the space constraints, we put the following figures and tables in the appendix.

- Fig. K.1: Garment transfer.
- Fig. K.2: Simultaneous try-on of multiple garments.
- Figs. K.3 and K.4: Prompts for GPT evaluator.

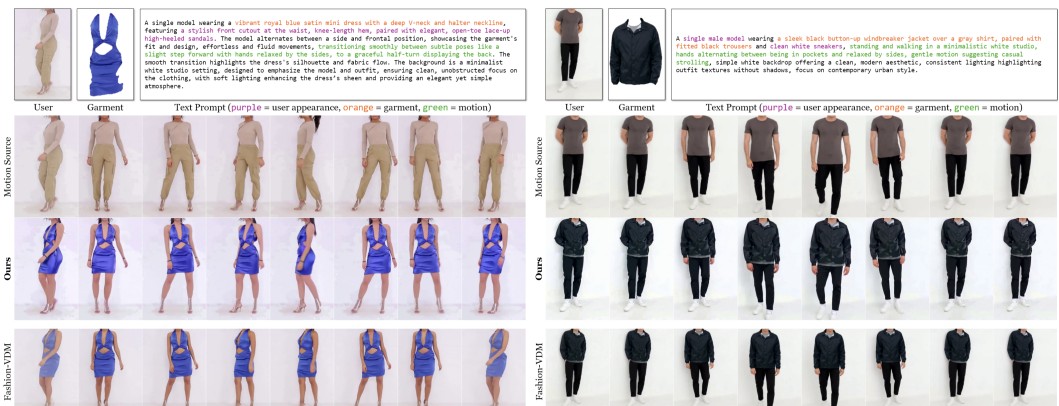

Figure K.1: Our **Dress**&**Dance** supports transferring a garment from another given image via segmentation, regardless of the pose of that image. Notably, our **Dress**&**Dance** generates high-resolution videos at $1152 \times 720$ with clear appearance and more details, while the baseline Fashion-VDM (Karras et al., 2024) exhibits color fading and generates low-resolution $512 \times 384$ videos.

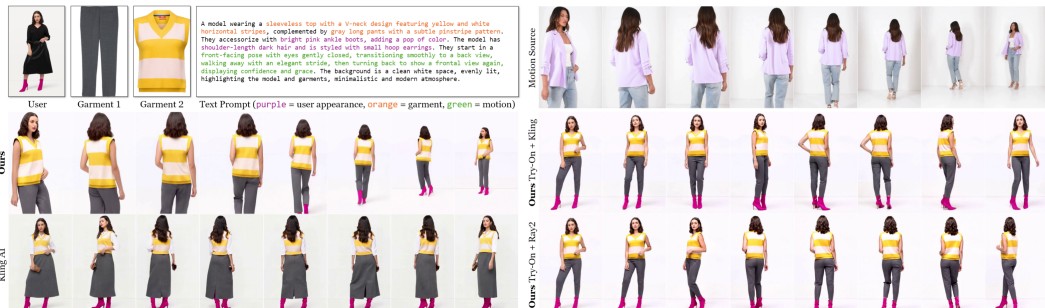

Figure K.2: Dress&Dance supports the simultaneous try-on of a top and bottom garment, correctly understanding and representing both without explicit labeling. In contrast, the Kling AI baseline (Kling, 2024) incorrectly misrepresents the trousers as a skirt.

```
You are a professional fashion expert and an evaluator of human video generation.
You task is to evaluate the overall quality of the generated human fashion video.
In each task, You will be given:
    (1) a user image, which is a photo of the model person;
    (2) one or two garment frontal images, which are photos of the garments to be worn by
    ↪   the model person
        (these images may be different from the user image, as they are the garments to be
        ↪   try-on)
        along with several image or video as conditions, with each of their descriptions;
    (3) a generation prompt, with is a detailed description of a human fashion video,
        you can assume that the prompt will be consistent with the model person and the
        ↪   garment images;

Then, you will be given one candidate video to evaluate, which is expected to generate
    the human fashion video of the model person wearing the garment(s) in the garment images,
    following the generation prompt.
To evaluate the quality of the human video, you should consider the following aspects:
- "user": user appearance fidelity
Whether the appearance of the person in the generated video matches the user image.
Also, for all the other garments that are not indicated to change in the prompt and garment
↪   images,
        the person should wear the same garments as in the user image.
- "garment": garment appearance fidelity
    Whether the appearance of the garments in the generated video matches the garment images,
        where the frontal of the garment should match the garment image from all the
        ↪   viewpoints at any human motion,
        and the back and sides of the garment, which are not shown in the garment images,
        ↪   should be reasonably generated
        to maintain a consistent appearance with the frontal.
- "motion": video motion quality
    Whether the motion of the person, garments, and possible accessories in the generated
    ↪   video is realistic, smooth, and natural,
        with clear appearance with reasonable speed, acceleration, and deceleration,
        without sudden or unnatural movements or morphing.
- "visual": visual quality
    Whether the generated video is visually appealing, with reasonable brightness, contrast,
    ↪   and color saturation,
        with smooth and natural motion, and without visual artifacts.
    The person, garments, and background should be visually clear and sharp, with high
    ↪   resolution and without blurring.
    The video should be visually appealing and engaging to watch.
- "overall": overall quality
    This is an "overall" evaluation, which should consider all the aspects of the video and
    ↪   try-on quality.
    Which means it should evaluate
        the fidelity of user image and garment images, the motion quality, and the visual
        ↪   quality,
        and give an overall score based on these aspects.
    To receive a high "overall" evaluation score, the video should satisfy all the aspects'
    ↪   requirements,
        and any failure in any aspect will result in a lower score.
    The overall score should be a comprehensive evaluation of the quality of such video
    ↪   try-on generation.
You will be paid tips based on the quality of your evaluation.
```

Figure K.3: GPT evaluator prompt: different aspects.

```
{{ For: each frame }}
This is the frame at {timestep} of the generated video.
{{ End For }}

Now, please evaluate the quality of the generated human video.
You should consider the four aspects: "user", "garment", "motion", and "visual", and finally
↪  provide an "overall" evaluation.
You should consider all the frames from the video, and have a comprehensive evaluation.
In your evaluation, for each scoring item of "user", "garment", "motion", "visual", and
↪  "overall",
    you should give an integer score from 0 to 100,
    where higher is better.
To better align your score:
    "0" means the worst, failure, or completely wrong;
    "60" means a passable quality, the generation is not contradict with the inputs, but
    ↪  with multiple noticeable issues;
    "80" means a good quality, with minor issues that can be ignored;
    "90" means a very good quality, with only very minor issues that are hard to notice;
    "100" means perfect and flawless quality, with no issues at all.
For "user" and "garment",
    a score of 80 means that the appearance of user or the frontal of garment is almost the
    ↪  same as the image.
For "motion",
    a score of 80 means that the motion is at least be able to be performed by a human,
    with no obvious artifacts like morphing, or the head turns 360 degrees.
For "visual",
    a score of 80 means that the video is visually clear and sharp, with only a few blurring,
    and the brightness, contrast, and color saturation are somehow reasonable.
For "overall",
    a score of 80 means that the video can roughly score 80 in all the other aspects above.
You can provide intermediate scores like 73, 86, 92, etc., to indicate the quality more
↪  precisely.
In your reply, you should provide the evaluation scores for each aspect in json format.
For example, the evaluation can be:
    {
        "user": 93,
        "garment": 87,
        "motion": 81,
        "visual": 62,
        "overall": 75,
    }
or
    {
        "user": 92,
        "garment": 64,
        "motion": 89,
        "visual": 55,
        "overall": 71,
    }
These are just examples. You should provide the actual evaluation for the actual video.
Please do not include any other contents, like quotes, or markdown code blocks like
↪  "```json".
You can just directly reply with the json string, starting with "{" and ending with "}".
All other information or incorrect format will be considered as invalid,
    which will result in a panelty in your payment.
Please provide your evaluation as a json string now:
```

Figure K.4: GPT evaluator prompt: rubrics.

