# OpenReview forum: "Dress&Dance: Dress up and Dance as You Like It"
_ICLR.cc/2026/Conference — Submitted to ICLR 2026_

### Official Review · Reviewer_bChG · 2025-10-20

**Soundness:** 2
**Presentation:** 3
**Contribution:** 3
**Rating:** 6
**Confidence:** 5

**Summary:**

This paper introduces Dress&Dance, a video diffusion framework enabling high-resolution virtual try-on videos, given a single user image, desired garment(s), and a reference motion video. The core innovation is the CondNets attention-based conditioning mechanism, designed to unify heterogeneous inputs—garments, users, motion references, and text—into a single token sequence suitable for scalable DiT-based video diffusion. The authors also address data scarcity and efficiency with a synthetic triplet data creation process and a multi-stage curriculum training strategy, facilitating training for large-scale video generation with limited paired try-on data. Extensive experiments are conducted with both quantitative and qualitative results, showing Dress&Dance outperforms strong baselines—including recent diffusion-based, commercial, and open-source methods—on metrics of fidelity and visual quality, and in support for complex motions and composite try-ons.

**Strengths:**

1. Scalable, Efficient Training Pipeline: The combination of synthetic triplet data creation (Section 3.2) and curriculum learning, where complexity increases along both task and resolution axes, is both pragmatic and generally applicable. The video-image hybrid training design (page 7) wisely utilizes abundant image data while aligning with the final inferential target.
2. Comprehensive, Robust Evaluation: The paper conducts both quantitative and qualitative comparisons, including commercial (Kling, Ray2) and recent strong open-source baselines, as well as ablative studies. Metrics span PSNR/SSIM/LPIPS and GPT4-based human/evaluator scoring, with cross-checks on Internet and captured datasets.
3. State-of-the-art Results and Strong Visuals: On core quantitative metrics, Dress&Dance surpasses all open-source and commercial baselines in garment and user appearance fidelity, and offers visually plausible, temporally consistent motion as attested by the sequences in the supplementary videos.
4. Practical System Contributions: Fast inference, user-facing modes, text-only try-on, and garment transfer (including multiple garments without explicit label) illustrate the system’s practicality and flexibility.

**Weaknesses:**

1. Overstated Claims and Weakness in Theoretical/Algorithmic Rigor: Much of the novelty is architectural and procedural, not fundamentally theoretical; e.g., Section 3’s CondNets is a nontrivial compositional application of attention blocks (as detailed in Figure 4 and G.1), but does not introduce fundamentally new attention or conditioning mechanisms—aware reviewers may view this as strong engineering and system design, not a conceptual breakthrough. E.g., equations governing training loss, attention fusion, and effect of garment-aware steering are described informally, and there are no explicit equations for the loss/objective (other than hints in the architecture description).
2. Lack of Ablation Study of CondNets: I think just encoding all the condition images (garment image, user image, skeleton images) into tokens and then input them into the DiT architecture should be able to deal with the try-on task (the subtask of video editing). The paper should include more explanation of the novelty of the proposed CondNets and add correponding ablation study.

**Questions:**

1. The fast inference speed (5-second clips at 24FPS and 1152×720resolution) and strong visual results come from the engineering techniques and rather than the proposed CondNets. This paper reads more like a technical report.
2. To foster further advancement in video try-on, it is hoped that the authors could open-source their dataset.
3. The details of CFG distillation. Necessary fomulas and related works should be included the appendix.

---

> ### Author Response · Authors · 2025-11-26
> **Response to Reviewer bChG (1/2)**
>
> We sincerely thank the reviewer for the constructive feedback.
>
> > Overstated Claims and Weakness in Theoretical/Algorithmic Rigor: Much of the novelty is architectural and procedural, not fundamentally theoretical; e.g., Section 3’s CondNets is a nontrivial compositional application of attention blocks...
>
> * We appreciate the reviewer recognizing our method as a “nontrivial compositional application” of attention blocks. To address the concern about theoretical or algorithmic novelty, we would like to clarify that our contribution does not aim to introduce a new attention operator, but rather to develop a **conditioning architecture that unlocks capabilities previously infeasible in the virtual try-on domain** – specifically, generating high-quality try-on videos of users performing complex motions.
>
> * This type of **architectural contribution** is well-established and highly valued in top-tier venues. For example, ControlNet (ICCV 2023 Best Paper) did not propose new attention or conditioning mechanisms; instead, its impact came from a novel architectural design that enabled new forms of controllable generation. In a similar spirit, our CondNets introduce a **more general and more flexible conditioning framework**, tailored for multi-modal virtual try-on but also potentially generalizable to other transformer-based models. Crucially, CondNets extend the controllability paradigm beyond what ControlNet-style architectures can support:
>
>     * ControlNet is inherently limited to supporting **pixel-aligned** conditions (e.g., Canny edges or depth maps that spatially match the output).
>     * In contrast, CondNets leverage the attention mechanism to fuse complex, **non-pixel-aligned** conditions (e.g., mapping a static, flat garment image to a dynamic, moving user in a video). This generalization extends the applicability of control mechanisms beyond the spatial constraints of previous works like ControlNet.
>
> > Lack of Ablation Study of CondNets: I think just encoding all the condition images (garment image, user image, skeleton images) into tokens and then input them into the DiT architecture should be able to deal with the try-on task (the subtask of video editing). The paper should include more explanation of the novelty of the proposed CondNets and add correponding ablation study.
>
> * We would like to respectfully clarify that the specific mechanism of "inputting tokens into the DiT architecture" effectively is precisely the core challenge that CondNets aim to solve. A simple encoding approach is insufficient due to the deep, hierarchical nature of the generative backbone (CogVideoX):
>     * **Feature Space Mismatch:** In a deep Diffusion Transformer, the feature space evolves significantly across different blocks (from low-level signals to high-level semantics). A standard approach that simply encodes conditions into a fixed set of tokens (e.g., via a single MLP) produces a static representation.
>     * **Failure of Static Injection:** Feeding these same static tokens into every transformer block fails because they cannot align with the unique, evolving feature distributions of each layer. This prevents the attention mechanism from effectively connecting the visual conditions with the video latents at different depths.
>
> * Our CondNet addresses this by mirroring the backbone architecture to perform **dynamic feature space conversion**:
>     * Because CondNet shares the same architectural depth, it naturally transforms the condition features layer-by-layer.
>     * This ensures that at any given block $i$, the conditional features are already transformed into the specific feature space required by block $i$ of the main backbone. This layer-wise alignment is critical for the model to "understand" the conditions, which a simple encoder cannot achieve.
>
> * Furthermore, regarding the pursuit of simplicity, we respectfully highlight that CondNets actually achieve **structural simplicity** compared to standard try-on methods:
>     * Prior methods typically necessitated complex, ad-hoc engineering, such as designing specialized warping modules or attention layers for garments and separate, distinct encoders/adapters for pose.
>     * In contrast, our framework unifies all heterogeneous conditions under a **single, homogeneous architectural template**. By using only one simple, repeated structure to handle everything from text to images to videos, CondNet eliminates the need for condition-specific sub-networks, offering a streamlined and scalable solution that is **significantly less complex** than the highly heterogeneous architectures of previous works.

---

> > ### Author Response · Authors · 2025-11-26
> > **Response to Reviewer bChG (2/2)**
> >
> > > The fast inference speed (5-second clips at 24FPS and 1152×720resolution) and strong visual results come from the engineering techniques and rather than the proposed CondNets. This paper reads more like a technical report.
> >
> > * We would like to first clarify a terminological misunderstanding in the review. The metrics mentioned ("5-second clips at 24FPS and 1152×720 resolution") describe the **properties** of the generated video content, not the **inference speed** (latency). Producing high-resolution, high-frame-rate outputs is a capability of the model architecture itself rather than merely an engineering optimization for runtime speed.
> >
> > * We respectfully disagree with the view that the results stem solely from "engineering techniques."
> >     * As acknowledged by the reviewer and clarified above, CondNet is our core technical contribution. In fact, CondNet is not a peripheral optimization; it serves as the foundational backbone of our model. Without the CondNet architecture, it would be structurally impossible to integrate and align the heterogeneous modalities (video, text, garment image, user image, pose) required for this task.
> >     * Therefore, the "strong visual results" are a direct consequence of this architectural contribution, which enables the effective fusion of these complex signals.
> >
> > * Furthermore, our contributions extend beyond the architecture. As demonstrated in our ablation studies, the final performance is a result of the holistic system, including the novel multi-stage training curriculum and the garment-aware target steering guidance. These are methodological innovations designed to solve specific learning difficulties, rather than standard engineering tweaks.
> >
> > * We have clarified these points in the revision to avoid confusion between architectural contributions and implementation details, and to better highlight how the proposed CondNet framework enables the capabilities demonstrated in the paper.
> >
> >
> >
> > > The details of CFG distillation. Necessary fomulas and related works should be included the appendix.
> >
> > * We appreciate the reviewer's suggestion to improve the completeness of our technical description.
> >     * In the revised **Appendix Sec. I.2**, we have provided the explicit mathematical formulation for the CFG distillation used in our training.
> >     * We have also expanded the discussion to include relevant prior works, ensuring that the theoretical background and implementation details are fully documented.

---

> > > ### Comment · Reviewer_bChG · 2025-11-26
> > >
> > > Thanks for authors' response. The visualization performance is excellent and I hope a point that the engineer techniques are useful. However, the dataset is not released and it is hard for researchers to follow. Considering the above reasons, I will maintain the score of 6 but not raise it to 8.

---

> > > > ### Author Response · Authors · 2025-11-27
> > > >
> > > > We sincerely thank the reviewer for the prompt follow-up and for appreciating the visualization quality and the utility of our technical approach.
> > > >
> > > > We fully understand the concern regarding dataset accessibility for future research. To address this while ensuring strict adherence to data privacy and copyright standards, our dataset will be hosted by a third-party vendor. Access will be granted subject to due diligence and proper licensing agreements. This protocol is established to guarantee the responsible and proper use of the human data involved.
> > > >
> > > > In addition, we hope to further facilitate reproducibility through the resources mentioned in our reproducibility statement:
> > > > 1.  **Codebase & Benchmarks:** We will release the code and the benchmark suite (along with our raw video outputs) to enable direct reference and fair comparison.
> > > > 2.  **Documentation:** We have documented detailed implementation details in the Appendices (Secs. G, F, H, and I).
> > > >
> > > > We hope that these resources — code, accessible data, and detailed documentation — will make our work easy to follow and build upon for the community. We hope that this clarification regarding data access and reproducibility effectively addresses the reviewer’s remaining concern.

---

### Official Review · Reviewer_wxd2 · 2025-10-28

**Soundness:** 3
**Presentation:** 3
**Contribution:** 2
**Rating:** 4
**Confidence:** 3

**Summary:**

This paper introduces Dress&Dance, a video diffusion framework that generates high-resolution (1152×720, 24 FPS) virtual try-on videos, allowing users to visualize various garments during complex motions.
The proposed CondNets architecture effectively integrates multi-modal inputs, while garment-aware target steering ensures precise garment placement.
Additionally, a synthetic triplet generation and multi-stage training strategy address data and computational challenges.
Experimental results demonstrate that Dress&Dance surpasses existing open-source and commercial solutions in garment detail preservation, user identity, and motion fidelity.

**Strengths:**

- The paper is clearly written and easy to follow.
- The proposed scenario is novel and highly relevant to real-world applications.
- The three-phase Garment-Aware Target Steering Guidance is innovative and experimentally validated to be effective.
- Experimental results demonstrate that Dress&Dance achieves state-of-the-art performance, outperforming existing methods.

**Weaknesses:**

- The proposed scenario requires solving two key tasks: garment transfer and motion synthesis. There are two straightforward baseline approaches: (1) performing garment transfer on the reference image followed by motion transfer; (2) applying motion transfer first and then garment transfer on the resulting video. The authors only compare with the first approach and do not provide analysis or comparison with the second approach.
- The proposed method employs text as a guiding condition. However, in real-world applications, users typically only provide a reference image, garment image, and motion video, without any textual input. Many pose-guided generation tasks do not consider text conditions; the rationale for including text guidance in this work is unclear.
- The framework diagram in Figure 4 is confusing, whereas Figure G.1 is much easier to understand. Since each condition is encoded by forwarding through a model with the same architecture as the main diffusion model, there are concerns about the efficiency of this encoding approach during both training and inference.
- The authors claim that CondNets provide a unified approach for encoding multi-modal control signals, but it is unclear why such a unified encoding is necessary for this specific task. The motivation for this design is not well-justified, nor is its extensibility experimentally validated. Using separate, lightweight encoders for different conditions could be more efficient, especially for conditions with lower complexity.
- In Figure 4, the reason for applying VAE encoding to the noise input and the role of the conditional output are not clearly explained.
- The multi-stage training curriculum is not novel, as it is a standard approach in training multi-resolution generative models.

**Questions:**

See Weaknesses.

---

> ### Author Response · Authors · 2025-11-26
> **Response to Reviewer wxd2 (1/3)**
>
> We sincerely thank the reviewer for the constructive feedback.
>
> > The proposed scenario requires solving two key tasks: garment transfer and motion synthesis. There are two straightforward baseline approaches: (1) performing garment transfer on the reference image followed by motion transfer; (2) applying motion transfer first and then garment transfer on the resulting video. The authors only compare with the first approach and do not provide analysis or comparison with the second approach.
>
> * We respectfully clarify that while both approaches are conceptually valid, they differ fundamentally in task complexity and technical maturity.
>     * Approach (1) decomposes the problem into "Image Try-On" and "Image-to-Video Generation (a.k.a. Motion Transfer)," both of which are relatively well-established image-based tasks. This effectively reduces the complexity, making it a suitable baseline.
>     * In contrast, Approach (2) decomposes the problem into "Image-to-Video Generation (a.k.a. Motion Transfer)" and **"Video-to-Video Try-On."** Unlike the first approach, this second stage of video-to-video try-on does not decrease the task difficulty; rather, it presents the exact same challenges of video try-on (e.g., garment and user appearance fidelity and consistency, motion smoothness, etc.) as our main task.
>     * Consequently, Approach (2) is not a simplified baseline like Approach (1), but rather a pipeline dependent on another high-difficulty video generation task.
>
> * Furthermore, our evaluation in **Fig. 7** effectively covers the core performance capability of Approach (2).
>     * In Fig. 7, we compare our method against state-of-the-art video-to-video try-on methods (e.g., SwiftTry, GPD-VVTO). This comparison can be viewed as evaluating the second stage of Approach (2) under the assumption of "perfect motion transfer" (since these methods take real video as input, instead of a video produced by motion transfer).
>     * Our results show that even with this optimal input, these methods struggle to achieve the high resolution and temporal consistency that our method provides. Constructing a full Approach (2) pipeline with actual (imperfect) motion transfer algorithms would only introduce additional artifacts and resolution bottlenecks (as most motion transfer models operate at lower resolutions), leading to strictly worse performance than the results we have already reported.
>
>
> > The proposed method employs text as a guiding condition. However, in real-world applications, users typically only provide a reference image, garment image, and motion video, without any textual input. Many pose-guided generation tasks do not consider text conditions; the rationale for including text guidance in this work is unclear.
>
> * We clarify that including text guidance does not impose an additional burden on the user, but rather enhances the system's robustness and flexibility.
>     * In practical applications, users do not need to manually provide text. As detailed in Appendix Sec. F.1, the required text prompts can be automatically derived from the input images using Vision-Language Models (VLMs). In fact, all experiments in our paper utilized VLM-generated prompts, demonstrating that our framework operates seamlessly in a standard image-input workflow.
>
> * Furthermore, the text condition serves critical functions that visual inputs alone cannot easily fulfill:
>     * **Handling Missing Information:** Text serves as a flexible descriptive tool for regions not covered by the reference images. For example, if a user currently wearing a full-body dress wants to try on only a top, a visual-only model lacks information about what bottom to generate. In our framework, the text condition (e.g., "wearing denim jeans") can resolve this ambiguity without requiring the user to upload an extra image of trousers, as also recognized by Reviewer Pvbd.
>     * **Text-Based Control:** As demonstrated in Fig. F.1 and recognized by Reviewer bChG, the text interface enables "text-guided try-on capabilities." This allows the model to not only perform the try-on but also adjust specific garment attributes or harmonize the non-try-on regions based on semantic descriptions, offering a level of outfit-level control that purely pose- and image-guided methods cannot support.

---

> > ### Author Response · Authors · 2025-11-26
> > **Response to Reviewer wxd2 (2/3)**
> >
> > > (Concerns regarding the clarity of Fig. 4 and the efficiency of the CondNet architecture.)
> >
> > * We appreciate the feedback regarding the figures and clarify their distinct roles:
> >     * **Figure 4** highlights the **key design and core mechanism** of our method. It zooms in on the specific interaction logic within the transformer blocks to illustrate how multi-modal signals are injected and fused via attention.
> >     * **Figure G.1**, on the other hand, presents the **comprehensive implementation details** and the holistic network structure. It depicts the full stack of layers, the integration of LoRA adapters, and the auto-regressive pipeline.
> >     * We intentionally separated them because merging the dense implementation details of Fig. G.1 into the conceptual view of Fig. 4 would obscure the key design principles with excessive clutter. However, we have revised Figure 4 in the updated paper to improve its clarity.
> >
> > * Regarding the efficiency concern, we respectfully clarify that employing a similar architecture for condition encoding implies only a marginal increase in computational cost due to the quadratic complexity of attention mechanisms ($O(N^2)$ for an attention sequence of $N$ tokens). To illustrate this, we provide the specific token lengths used in our model: $L_{\mathrm{video}} = 35,640$, while condition lengths are significantly shorter ($L_{\mathrm{skeleton}} \approx 5,000$, $L_{\mathrm{image}} = 3,240$, $L_{\mathrm{text}} = 226$).
> >     * **Main Backbone Dominance:** The self-attention computation scales with the square of the total concatenated sequence length: $(L_{\text{video}} + L_{\text{text}} + \sum_{i} L_{\text{cond}, i})^2$. Since $L_{\text{video}}$ (35,640) is an order of magnitude larger than even the largest condition sequence, the computational cost is overwhelmingly dominated by the video tokens: $(L_{\text{video}} + \dots)^2 \approx L_{\text{video}}^2$.
> >     * **Marginal CondNet Cost:** In contrast, each CondNet processes only its specific condition tokens independently. For example, a garment CondNet operates on $(L_{\text{text}} + L_{\text{image}})^2 \approx (226 + 3,240)^2$.
> >     * **Mathematical Comparison:** Because $L_{\text{video}}^2 \gg (L_{\text{text}} + L_{\text{image}})^2$ (i.e., $35,640^2 \approx 1.27 \times 10^9$ vs. $3,466^2 \approx 1.2 \times 10^7$), the computational load of each CondNet branch is negligible (approximately 1\%) compared to the main backbone.
> >     * Furthermore, as detailed in Appendix Sec. I, we have implemented acceleration techniques that further optimize the inference speed, ensuring the model remains efficient for practical deployment.
> >
> > > (Questions regarding the necessity and justification of the CondNet design compared to separate lightweight encoders.)
> >
> > * We would like to respectfully clarify the distinct roles within our framework to address the concern regarding efficiency and necessity.
> >     * First, "CondNets" function *not* as a raw signal encoder, but as a **conditioner architecture** responsible for injecting control signals into the generative process. The actual raw encoding of inputs (e.g., images) is indeed handled by standard, lightweight VAE encoders (indicated by the black arrows in Fig. 4 and Appendix Fig. G.1).
> >     * Second, the motivation for employing a mirrored architecture in CondNets is to ensure precise **feature space alignment**. The main Diffusion Transformer (DiT) evolves its feature representation through a deep stack of attention blocks, creating a highly complex and specific feature space within each attention block.
> >
> > * Regarding the suggestion to use separate, simpler conditioners (“encoders”), we argue that such methods are insufficient for this task.
> >     * A simple or shallow network lacks the structural capacity to track the complex, non-linear transformations occurring within the deep attention layers of the main backbone.
> >     * By reusing the DiT architecture, CondNets naturally align the heterogeneous conditional features to the specific feature space of each corresponding transformer block in the main model. This design is critical for "homogenizing" the conditions, ensuring that the control signals remain semantically and spatially aligned with the generated video latent features throughout the entire generation process — an alignment that simple conditioners (“encoders”) typically fail to learn effectively.
> >     * In fact, compared to prior try-on methods, our CondNet design represents a step towards unification and simplicity. Previous image and video try-on approaches (e.g., TPD, OOTD, ViViD, Fashion-VDM) typically necessitate specialized conditioning modules or distinct architectures tailored for each specific condition type (e.g., specific warping modules or attention layers for garments versus different encoders for pose). In contrast, our framework unifies all conditions under a single and simple architectural template, eliminating the need for such ad-hoc, condition-specific engineering.

---

> > > ### Author Response · Authors · 2025-11-26
> > > **Response to Reviewer wxd2 (3/3)**
> > >
> > > > In Figure 4, the reason for applying VAE encoding to the noise input and the role of the conditional output are not clearly explained.
> > >
> > > * We apologize for the confusion and would like to clarify a misunderstanding regarding the visualization in Fig. 4. The VAE encoder is not applied to the noisy video directly. Instead, consistent with the standard Latent Diffusion framework, the ground truth video is first encoded into latent space, and Gaussian noise is subsequently added to these latents to obtain the noisy input.
> > >
> > > * Regarding the role of the conditional output for each transformer block, it functions as the recursive input for the subsequent block. Similar to how the main video and text features are processed, the conditional features are updated and passed forward layer-by-layer. This ensures that the control information evolves in synchronization with the generation process throughout the network depth.
> > >
> > > * To prevent further confusion, we have revised **Fig. 4** and **Fig. G.1**. We removed the potentially misleading VAE encoder depiction near the noise input and updated the captions to explicitly state that the sequence outputs from one block serve as the inputs for the next.
> > >
> > > > The multi-stage training curriculum is not novel, as it is a standard approach in training multi-resolution generative models.
> > >
> > > * We respectfully clarify that while stage-wise training is indeed a common concept, our specific strategy addresses a fundamentally different challenge compared to standard multi-resolution generative models (e.g., **Fashion-VDM**):
> > >     * Standard methods typically train new models from scratch, utilizing progressive spatial and temporal scaling to gradually build generation capabilities from the ground up.
> > >     * In contrast, our approach builds upon the pre-trained CogVideoX-5B. The core challenge here is to inject specific try-on control signals **without suffering from catastrophic forgetting of the foundation model's robust spatial and temporal priors** (e.g., text understanding and long-term consistency).
> > >
> > > * We emphasize that the standard scaling mechanisms used in previous works are not directly applicable to our setting, as they would **disrupt the learned feature space and compromise these pre-trained capabilities**. To address this, we designed a specific video-image hybrid training strategy:
> > >     * We combine low-resolution but full-length videos with full-resolution images (as detailed in L369).
> > >     * This unique design allows for fast iteration while **preserving** the model’s pre-trained capability to generate "high-resolution" and "41-frame" videos, a balance that traditional scaling strategies fail to achieve in a fine-tuning context.
> > >
> > > * Furthermore, our curriculum is specifically tailored to our unique **CondNet architecture**:
> > >     * Since CondNet mirrors the deep architecture of the main DiT to ensure feature alignment, it requires a specialized training schedule to converge effectively. The proposed curriculum is not a generic application of multi-stage training but a necessary design choice to enable the successful training of this specific architectural innovation.
> > >
> > > * **Paper Revision:** To highlight this distinction, we have added a detailed discussion in **Sec. 3.2** of the revised paper. We explicitly differentiate our approach from standard progressive scaling methods to justify the necessity and novelty of our hybrid strategy for fine-tuning foundation models.

---

### Official Review · Reviewer_Pvbd · 2025-10-28

**Soundness:** 3
**Presentation:** 2
**Contribution:** 3
**Rating:** 4
**Confidence:** 5

**Summary:**

Dress&Dance focuses on generating virtual try-on videos given a person image, garment image(s), video showcasing the desired motion, and optionally text. This paper sets out to address the challenges of (1) high-quality input person/garment fidelity, (2) fine-grain motion preservation from input video, and (3) challenging multi-modal, non-pixel-aligned inputs.

Their contributions include:
- SOTA performance on synthesizing videos up to 5s at 1152x720px resolution.
- CondNets: An attention-based conditioning architecture to unify multimodal inputs into a single token sequence.
- Garment-aware target steering: A guidance mechanism to better balance text conditioning with other conditionings
- Synthetic triplet generation: Leveraging pretrained models to generate paired training data.
- Multi-stage training curriculum: Progressively increasing frame rate and spatial resolution during training

**Strengths:**

- The video results are compelling and clearly preserve sharp garment details.
- The authors are obviously well-versed in recent literature, and leverage many state-of-the-art strategies (e.g. cfg distillation, LoRA finetuning, etc.) and models (e.g. DiT) for their method.
- Extensive SOTA comparisons, including to recent open-source, commercial models, such as Kling Video 1.6 and and Ray2. These show the superiority of Dress&Dance for detail preservation and motion fidelity.
- Although the methods are not the most novel in my opinion, the overall framework achieves state-of-the-art results in video virtual try-on, which is a laudable contribution.

**Weaknesses:**

- Given similar strategies in numerous papers, such as Matryoshka Diffusion Models and Fashion-VDM, I suggest scaling down claims that multi-stage training curriculum a novel contribution.
- The Method section was missing many key details (see Questions below)
- Classifier-free guidance distillation training is not new, it is utilized in many popular SOTA models, including Flux. See https://arxiv.org/pdf/2210.03142. Missing citation for this or explanation of why it is different.
- The figures/tables could use some work. Specifically, The fonts in most of the figures (e.g. Fig. 4) are quite small. Additionally, Fig. 5 is very, very small and it is hard to see the frames.
- The ethics statement does not mention consent, compensation, or conditions provided by/for the 183 persons hired for their dataset generation.
- At ~1:21 of the video, I observe the buttons re-appearing in the back of the garment. This Janus problem is not mentioned as a limitation of the method.

**Questions:**

1. Lines 239-241: Along which dimension are the conditional sequences concatenated with the video/text sequences?
2. How does CondNets differ from past methods that finetune DiT with LoRA for new conditioning inputs? For example, EasyControl, IC-LoRA, various Flux pipelines, etc.
3. Are the VAE encoders fine-tuned or frozen for the new conditioning inputs?
4. Garment-Aware Target Steering Guidance: Why do phase 1 and 2 need to be the layflat garment? Why not just gray-out the non-garment parts of the target frames with decreasing opacity for phases 1-3?
5. 299-300: Which pretrained model(s) is used for synthetic triplet generation? What were the prompts?
6. How are bottoms handled for top try-on (or tops for bottom try-on)? I notice in the video that the bottom is not preserved.

**Details Of Ethics Concerns:**

1. The paper (without references) is over the 9 page limit.
2. This paper mentions hiring 183 human models to try on 100 garment sets. Though, no mention of consent, compensation, or working conditions is included.

---

> ### Author Response · Authors · 2025-11-26
> **Response to Reviewer Pvbd (1/3)**
>
> We sincerely thank the reviewer for the constructive feedback.
>
> > Given similar strategies in numerous papers, such as Matryoshka Diffusion Models and Fashion-VDM, I suggest scaling down claims that multi-stage training curriculum a novel contribution.
>
> * We would like to respectfully clarify that our training strategy addresses a fundamentally different challenge compared to methods like Matryoshka Diffusion Models and Fashion-VDM:
>     * Those methods typically train new models from scratch, relying on progressive spatial and temporal scaling to gradually learn generation capabilities from the ground up.
>     * In contrast, our approach builds upon the pre-trained CogVideoX-5B. The core challenge here is to inject specific try-on control signals **without suffering from catastrophic forgetting of the foundation model's robust spatial and temporal priors** (e.g., text understanding and long-term consistency).
>
> * We emphasize that the standard scaling mechanisms used in previous works are not applicable to our setting, as they would **disrupt the learned feature space and compromise these pre-trained capabilities**. To address this, we designed a specific video-image hybrid training strategy:
>     * We combine low-resolution but full-length videos with full-resolution images (as detailed in L369).
>     * This design allows for fast iteration while **preserving** the model’s pre-trained capability to generate "high-resolution" and "41-frame" videos, a balance that traditional scaling strategies fail to achieve in a fine-tuning context.
>
> * To clarify this distinction, we have added a detailed discussion in the revised paper. As shown in **Sec. 3.2**, we explicitly differentiate our curriculum design from previous from-scratch training strategies to justify its necessity and novelty in this specific context.
>
> > Classifier-free guidance distillation training is not new, it is utilized in many popular SOTA models, including Flux. Missing citation for this or explanation of why it is different.
>
> * We thank the reviewer for pointing out this relevant reference. We have included the citation in the revised version (Sec. 3.2, Appendix Sec. I.2) and added a discussion acknowledging the foundational role of prior CFG distillation works.
>
> * Regarding the technical distinction, we clarify that while our approach builds upon the established concept of CFG distillation, we introduce specific adaptations to handle multiple heterogeneous conditions simultaneously:
>     * Unlike standard distillation applied to single-condition models (e.g., text-only), our framework must balance conflicting guidance requirements from text, human pose, and garment reference images.
>     * Our strategy specifically addresses cases where these conditions require distinct guidance strengths. By adapting the distillation objective to this multi-condition setting, we ensure that high guidance fidelity is maintained across all modalities without one dominating or suppressing the others — a challenge not addressed in the original single-condition formulation.
>
> > The figures/tables could use some work. Specifically, The fonts in most of the figures (e.g. Fig. 4) are quite small. Additionally, Fig. 5 is very, very small and it is hard to see the frames.
>
> * We apologize for the readability issues in the initial submission. We have taken the advice to heart and extensively revised the visualization layout.
>
> * In the updated paper, we have enlarged the font sizes and increased the scale of the figures, particularly **Fig. 4** and **Fig. 5**. These adjustments ensure that all frames, details, and text annotations are now clearly visible and legible without zooming in.
>
> > The ethics statement does not mention consent, compensation, or conditions provided by/for the 183 persons hired for their dataset generation.
>
> * We take ethical considerations very seriously and have updated the Ethics Statement in the revised paper to explicitly cover these points:
>     * **Conditions:** The data collection was conducted in a fully equipped professional studio environment. To ensure the safety and comfort of the participants, each subject was assisted by a dedicated support team of five professionals, including photographers and stylists.
>     * **Consent:** We confirm that informed consent was obtained from all participants. Each subject signed a formal service agreement prior to the session, explicitly consenting to the data capture and its usage for research purposes.
>     * **Compensation:** All subjects were fairly and generously compensated for their time and effort. The compensation rates provided were highly competitive (comparable to the hourly rates of senior engineering roles), ensuring the ethical treatment of all participants involved.

---

> > ### Author Response · Authors · 2025-11-26
> > **Response to Reviewer Pvbd (2/3)**
> >
> > > At ~1:21 of the video, I observe the buttons re-appearing in the back of the garment. This Janus problem is not mentioned as a limitation of the method.
> >
> > * We would like to respectfully clarify that this phenomenon is distinct from the "Janus problem" (which typically refers to geometric confusion where the model incorrectly renders a frontal face or body structure on the back).
> >     * Our model incorporates a modified ViTPose input that explicitly encodes frontal and dorsal information by **applying distinct color encodings to the left and right keypoints and limbs**. This explicit directional cue ensures the model is fully aware of the subject's orientation and geometry. More details are provided in Appendix Sec. G.4.
> >
> > * The phenomenon observed at ~1:21 is instead a result of the generative inference inherent to single-view try-on tasks:
> >     * Since the back of the garment is not visible in the single frontal reference image, the model must predict the back appearance based on the frontal visual cues to complete the view.
> >     * In this specific case, the model aimed to maintain stylistic consistency by generating a vertical stripe pattern on the back that aligns with the frontal design. While this pattern visually resembles the front details, it represents a reasonable texture prediction based on learned priors, rather than a geometric failure to distinguish front from back.
> >     * Furthermore, we emphasize that this specific phenomenon is an isolated instance. Extensive results in our supplementary video and figures demonstrate that for the vast majority of cases, the model correctly renders appropriate back textures according to, instead of replicating, the frontal elements, confirming that our method does not suffer from systematic geometric confusion.
> >
> >
> > > The paper (without references) is over the 9 page limit.
> >
> > * We respectfully clarify that our submission is fully compliant with the ICLR page limit policy. According to the official author guidelines, the "Ethics Statement" and "Reproducibility Statement" are explicitly excluded from the 9-page content limit. When these sections are excluded, our main text falls strictly within the required page limit.
> >
> > > Lines 239-241: Along which dimension are the conditional sequences concatenated with the video/text sequences?
> >
> > * We clarify that the sequences are concatenated along the **sequence dimension**.
> >     * In terms of tensor representation, if the input sequences have the shape $[B, L, C]$ (where $B$ is batch size, $L$ is sequence length, and $C$ is channel dimension).
> >     * Concatenating multiple sequences with lengths $L_1, L_2, L_3$ along this dimension results in a combined tensor of shape $[B, L_1 + L_2 + L_3, C]$. This allows the model to attend to all conditions jointly within the same attention mechanism.
> >
> > > How does CondNets differ from past methods that finetune DiT with LoRA for new conditioning inputs? For example, EasyControl, IC-LoRA, various Flux pipelines, etc.
> >
> > * We thank the reviewer for pointing out these relevant *concurrent* works. While our method shares the goal of efficient conditioning, there are significant differences in both scope and the interaction mechanism between text and conditions.
> >     * First, regarding the scope, most mentioned works focus primarily on single-modality image generation. In contrast, our CondNets are designed for a complex video virtual try-on setting, supporting simultaneous conditioning on multiple heterogeneous modalities (videos, images, sparse poses) and roles within a unified framework.
> >     * Second, and most importantly, the core architectural difference lies in how the conditions interact with the textual context. In these concurrent approaches, the conditional inputs are processed relatively independently (e.g., via *self-attention* layers within the condition branch) and do not directly interact with the text prompts.
> >
> > * In contrast, as illustrated in Fig. 4, our design explicitly enables *cross-attention* between the text prompts and the conditional inputs. This allows the text embeddings to dynamically update based on the visual conditions, establishing a deep semantic connection. This mechanism enables the text to serve as a global controller, allowing us to handle complex try-on scenarios using natural language descriptions rather than relying on specialized CondNet variants constrained to specific garment types (top, bottom, or whole-body).
> >
> > > Are the VAE encoders fine-tuned or frozen for the new conditioning inputs?
> >
> > * The VAE encoders are frozen during our training process. This adheres to the standard training strategy for latent diffusion models, ensuring that the pre-trained latent space remains stable while we focus on optimizing the generative backbone and conditioning adapters.

---

> > > ### Author Response · Authors · 2025-11-26
> > > **Response to Reviewer Pvbd (3/3)**
> > >
> > > > Garment-Aware Target Steering Guidance: Why do phase 1 and 2 need to be the layflat garment? Why not just gray-out the non-garment parts of the target frames with decreasing opacity for phases 1-3?
> > >
> > > * We would like to respectfully explain that the "gray-out" strategy suggested by the reviewer **will fail**, because it overlooks the critical interaction between the **"Attention Dominance"** problem and the **geometric difficulty** of the task.
> > >     * First, as discussed in L249, our model inherits a powerful pre-trained prior where text tokens "receive dominantly high attention scores," tending to "overshadow other conditions." This creates an inherent resistance to learning new visual controls.
> > >     * Second, the specific task requires solving complex "garment placement" (L183), which involves bridging a huge geometric gap between the flat input image and the warped, posed garment in the video. The reviewer's "gray-out" strategy only simplifies the background pixels but does not reduce this fundamental geometric difficulty.
> > >     * Consequently, since the geometric correspondence remains too difficult to learn immediately, even with non-garment parts grayed out, the model will fail to overcome the attention dominance. Instead, it follows the path of least resistance described in L256: it "learns to minimize the loss by directly overfitting to the text prompt," generating a generic garment rather than preserving the specific input details.
> > >
> > >
> > > * Therefore, our curriculum is specifically designed to break this Attention Dominance by decomposing the difficulty:
> > >     * The goal of Phase 1 (using the static layflat garment as the target) is to present an "easy win." Since the target is identical to the input condition, the model can significantly reduce training loss by simply learning to "copy" the raw content from the garment image (L287).
> > >     * This step is crucial because it creates a direct incentive for the attention mechanism: **to minimize the loss on this copying task, the model’s attention computation is compelled to explicitly attend to the specific features and pixels of the garment image**, thereby successfully breaking the text-only generation pattern.
> > >     * Once this dependency is established and the Attention Dominance is overcome, Phases 2 and 3 gradually introduce the spatial alignment between the garment and the user and the warping from the flat garment to the posed body. Since the model is already conditioned to attend to the garment features, it can then focus on learning the necessary spatial transformations without reverting to ignoring the condition.
> > >
> > >
> > > > 299-300: Which pretrained model(s) is used for synthetic triplet generation? What were the prompts?
> > >
> > > * We specify that we utilized three distinct pre-trained virtual try-on models to generate the synthetic triplets: **ML-VTON**, **TPD**, and **OOTD**.
> > >
> > > * Regarding the inquiry about prompts, we clarify that these specific image-based virtual try-on methods operate without textual inputs. They are purely image-conditioned models that take a source user image and a garment image as inputs to synthesize the user wearing the target garment while preserving the original pose. Therefore, no text prompts were involved in this generation process.
> > >
> > > > How are bottoms handled for top try-on (or tops for bottom try-on)? I notice in the video that the bottom is not preserved.
> > >
> > > * We clarify that the appearance of the non-target regions (e.g., bottoms during a top try-on) is primarily governed by the **text description** provided to the model.
> > >     * **Preservation:** In general, if the original bottom aligns with the text description or if the prompt is generic regarding the bottom, our model preserves the original appearance. This preservation capability is demonstrated in multiple qualitative results, such as Figs. 3, 6, 7, and K.1, where the non-try-on garments remain consistent.
> > >     * **Text-Guided Modification:** However, if the text description explicitly specifies a different style, or if the model infers that a stylistic update is needed to match the new garment based on the prompt, the model will modify the non-target region accordingly (as analyzed in Fig. F.1).
> > >
> > > * Regarding the observation in the video where the bottom is not preserved, we respectfully request if the reviewer could kindly point out the specific timestamp or case? We would be grateful for this detail, as it would allow us to examine whether it is an instance of text-guided outfit harmonization or a potential failure case that requires further discussion.

---

### Official Review · Reviewer_1C4t · 2025-10-30

**Soundness:** 2
**Presentation:** 2
**Contribution:** 2
**Rating:** 2
**Confidence:** 5

**Summary:**

This paper proposes Dress&Dance, a video diffusion framework for video virtual try-on with motion guidance. It introduces CondNets for multi-modal input unification, garment-aware target steering to address text-prior dominance, and synthetic data/training curriculum to solve data scarcity and computational issues.

**Strengths:**

1. CondNets effectively integrates heterogeneous inputs (garment images, user images, motion videos, text) via attention-based tokenization, overcoming the limitations of pixel-aligned methods like ControlNet for unaligned garment data.

**Weaknesses:**

1. The authors did not provide quantitative comparisons with existing video virtual try-on methods, such as GPD-VVTO, ViViD, and Tunnel Try-on. Only qualitative comparison is not sufficient to demonstrate the superiority of Dress&Dance against these SOTA methods.
2. GPT-based evaluation lacks transparency and may be biased by prompt design; no user study validates practical usability.
3. Directly comparing Dress&Dance with the baseline methods may not be entirely fair, given that Dress&Dance benefits from a substantial amount of extra training data. This raises uncertainty regarding whether the observed improvements stem from the added data or the unique architecture of Dress&Dance. To ensure a thorough and impartial assessment, it is recommended that the authors re-evaluate their approach using publicly accessible datasets like VVT and ViViD.
4. Understanding the computational resources required to run and train this model is crucial. Given that each condition necessitates a DiT, it appears that the model demands substantial GPU memory.
5. Synthetic triplet generation relies on external image try-on models, introducing potential errors that propagate to training. This dependency also limits generalization if pre-trained models perform poorly on rare garment types.

**Questions:**

See above

---

> ### Author Response · Authors · 2025-11-26
> **Response to Reviewer 1C4t (1/2)**
>
> We sincerely thank the reviewer for the constructive feedback.
>
> > The authors did not provide quantitative comparisons with existing video virtual try-on methods, such as GPD-VVTO, ViViD, and Tunnel Try-on. Only qualitative comparison is not sufficient to demonstrate the superiority of Dress&Dance against these SOTA methods.
>
> * We clarify that our original submission prioritized quantitative comparisons with strong baselines (such as Kling) in Tab. 1 and Tab. E.1. We respectfully highlight that our method generates videos at a significantly higher resolution ($1152 \times 720$) and frame rate (24 FPS) compared to the mentioned methods (e.g., GPD-VVTO and ViViD), which typically operate at resolutions lower than $720 \times 540$ with lower FPS. This capability for high-fidelity generation intrinsically demonstrates the superiority of our approach.
>
> * Moreover, we initially prioritized qualitative comparisons because the benchmarks used by these methods rely on low-resolution inputs. These inputs are unsuitable for high-fidelity generation models like ours and the strong baseline Kling. Feeding low-resolution data into high-resolution models results in a lack of necessary detail and upscaling artifacts, which creates an unfair evaluation setting. On such low-quality benchmarks, low-resolution models might even appear artificially competitive simply because the ground truth is also low-resolution. Therefore, we utilized qualitative comparisons to provide a more intuitive and accurate demonstration of the generation quality gap.
>
> * To fully address the concern, we have provided comprehensive quantitative evaluations in the revised version:
>     * **GPT-Based Metrics:** In **Appendix Tab. E.2** (setup in **Appendix Sec. E.3**), we evaluated these methods using GPT-based evaluation metrics. The results demonstrate that our method achieves consistently better performance than all the suggested baselines. We also observe that low resolution can inadvertently mask artifacts in baseline outputs; despite this potential advantage for the baselines, our method still outperforms them in overall quality and consistency.
>     * **User Study:** In **Appendix Tab. E.3** (details in **Appendix Sec. E.4**), we conducted a user study comparing our method against these VVT baselines. The results show a preference rate exceeding 50% in favor of our method, further confirming its superiority in human perceptual evaluation.
>
> > GPT-based evaluation lacks transparency and may be biased by prompt design; no user study validates practical usability.
>
> * To ensure objective assessment, our original submission prioritized standard reference-based metrics (e.g., PSNR) on the Captured dataset in Tab. 1. These metrics evaluate the method against **ground truth** independently of prompt design or subjective bias, thereby ensuring the reliability of our experimental conclusions.
>
> * Regarding the choice of GPT-based evaluation over a user study in the initial submission, we adopted GPT primarily as a scalable proxy due to the inherent limitations of human evaluation. As discussed in works like **VQAScore (ECCV 2024)**, human evaluations are often **"expensive"** and **"difficult to reproduce,"** making them less suitable for standardized benchmarking compared to automated metrics.
>
> * To further address the concern regarding practical usability and potential bias, we have conducted a user study with 40 participants in the revised version. As reported in **Appendix Tab. E.3** and **Appendix Sec. E.4**, we invited 40 participants to evaluate the generated results based on weighted preference. The user study results align with GPT-based  quantitative metrics, further confirming the perceptual superiority of our method.

---

> > ### Author Response · Authors · 2025-11-26
> > **Response to Reviewer 1C4t (2/2)**
> >
> > > Directly comparing Dress&Dance with the baseline methods may not be entirely fair, given that Dress&Dance benefits from a substantial amount of extra training data. This raises uncertainty regarding whether the observed improvements stem from the added data or the unique architecture of Dress&Dance. To ensure a thorough and impartial assessment, it is recommended that the authors re-evaluate their approach using publicly accessible datasets like VVT and ViViD.
> >
> > * We respectfully clarify that the performance improvement does not stem solely from data scale:
> >     * First, commercial models like Kling and Ray2 are trained on significantly larger-scale datasets than ours, yet they perform worse on the video virtual try-on task.
> >     * Second, the baseline ML-VTON (L436) was trained on the exact same dataset as our method but failed to achieve comparable results.
> >     * Finally, our ablation studies, which train weaker variants on our full dataset, demonstrate that the dataset alone is insufficient to produce high-quality results without our methodology contributions. These comparisons isolate the effectiveness of our method design from data advantages.
> >
> > * Regarding the suggestion to re-evaluate (or re-train) on VVT and ViViD, we emphasize that **our work targets a new setting: High-Resolution, High-FPS Video Virtual Try-On**, which requires data quality that existing public datasets cannot provide:
> >     * As explicitly noted in the ViViD paper, older datasets like VVT suffer from "low resolution, limited variety of clothing types, and simple motions," which make them "not conducive for the model to learn clothing detail representations."
> >     * We respectfully highlight that **this limitation similarly applies to the ViViD dataset itself** ($832\times 624$, highly compressed) when compared to our high-fidelity standards ($1152 \times 720$, 24 FPS) established in our work. **Existing public datasets lack the necessary resolution and motion complexity for this new setting.** Therefore, collecting a high-quality dataset is a necessary and critical step to enable our research and unlock the new virtual try-on capabilities we demonstrate, rather than an unfair advantage.
> >
> >
> > > Understanding the computational resources required to run and train this model is crucial. Given that each condition necessitates a DiT, it appears that the model demands substantial GPU memory.
> >
> > * We would like to respectfully clarify a misunderstanding regarding the model architecture: as described in L239 and Fig. G.1, our method does **not** employ an independent DiT for each condition. Instead, we utilize a single shared DiT backbone. The specific conditions are handled via LoRA adapters, which introduce only a negligible increase in parameter count and memory usage compared to the base model.
> >
> > * Regarding specific computational requirements, our model is built upon CogVideoX-5B, resulting in a final size of approximately 6B parameters. It is fully capable of performing training and inference on a single GPU with 40GB VRAM. This demonstrates that the memory demand is manageable and standard for current high-fidelity video generation tasks.
> >
> > > Synthetic triplet generation relies on external image try-on models, introducing potential errors that propagate to training. This dependency also limits generalization if pre-trained models perform poorly on rare garment types.
> >
> > * We acknowledge that relying on external models may introduce potential noise. However, we consider this strategy **critical for scalability** and reducing reliance on expensive human annotation. By leveraging synthetic triplets, our framework can utilize massive amounts of unpaired video and image data, which is essential for training a generalizable video foundation model.
> >
> > * To mitigate the impact of potential errors from these external models, we employ a **mixed-model strategy**:
> >     * Instead of relying on a single source, we utilize three distinct image try-on models (TPD, OOTD, and ML-VTON) to generate the synthetic triplets.
> >     * Since these models exhibit different performance characteristics and error patterns, this diversity prevents our model from overfitting to any specific artifact or error mode, effectively acting as a form of **data regularization**.
> >
> >
> > * Regarding the concern about generalization, we emphasize that the synthetic generation actually enhances robustness by enabling the use of large-scale unpaired data.
> >     * As demonstrated qualitatively in Fig. 7 and the supplementary video, our method successfully generalizes to various rare garment types (e.g., semi-transparency) and complex scenarios.
> >     * This empirical evidence suggests that our model learns robust representations and is not limited by the specific weaknesses of individual pre-trained models used in data generation.

---

### Author Response · Authors · 2025-11-26
**Summary: Manuscript Revision**

We sincerely thank the reviewers for their constructive feedback and address their concerns in each response.

In the revised manuscript, we have made the following updates to address the concerns raised (with changes marked in **blue**):

* **Additional Quantitative Experiments & User Study (1C4t):** Added objective metric comparisons against VVT baselines (**Appendix Tab. E.2**) and reported results from a 40-participant user study (**Appendix Tab. E.3**).
* **Clarification of Novelty (Pvbd, bChG):** Expanded **Sec. 3.2** to differentiate our curriculum from standard progressive scaling (e.g., Matryoshka Diffusion) and clarified the architectural distinctiveness of CondNets compared to pixel-aligned methods like ControlNet.
* **Technical Details & Visualization (Pvbd, wxd2):** Added mathematical formulations for CFG distillation (**Appendix Sec. I.2**), detailed the modified ViTPose representation (**Appendix Sec. G.4**), and revised **Figure 4** to better illustrate the recursive data flow.
* **Ethical Statement (Pvbd):** Updated the Ethics Statement with participant details (consent, compensation, conditions).

---

### Meta-Review · Area_Chair_bcKd · 2026-01-06

**Summary:**

The reviewers raise broad concerns about evaluation, novelty, efficiency, and clarity. The original version lacks quantitative comparisons with existing video virtual try-on methods (e.g., GPD-VVTO, ViViD, Tunnel Try-on), relies on opaque GPT-based evaluation without user studies, and presents potentially unfair comparisons since Dress&Dance uses substantially more training data, making it unclear whether gains stem from data scale or architectural choices. Many of these points have been clarified in the rebuttal. The method’s computational cost and GPU memory demands were not initially analyzed despite requiring multiple DiTs, and the synthetic triplet generation depends on external try-on models, which may introduce errors, limit generalization, and weakens claims of novelty given similar multi-stage curricula in prior work. Several components are not novel or insufficiently justified, including classifier-free guidance distillation, multi-stage training, and the CondNets design, whose motivation, efficiency, and benefits over simpler encoders are unclear and lack ablation studies. Methodological details and theoretical rigor are missing or informal, figures are hard to read, and limitations such as visual artifacts (e.g., Janus effects) are not discussed. Additional concerns include missing ethical disclosures for dataset collection, incomplete baseline comparisons (e.g., alternative task orderings), unclear necessity of text guidance for real-world use, confusing framework diagrams, and unexplained design choices (e.g., VAE encoding of noise), collectively weakening the paper’s credibility and claims of contribution.

**Reviewer Concerns:**

I think the authors did a good job in their rebuttal but they also make comments that are impossible to verify. For example, while I can see the comments of Reviewer bChG explicitly confirming that all concerns aside from dataset accessibility were resolved, I cannot see in the system any reply from Reviewer wxd2 (I can see that there was a comment but this has been deleted). I have no reason to doubt what the authors are claiming, I'm simply stating what I see and unfortunately I need to base my decision based on what I have. I do not think the concerns regarding the database are critical so I will not consider those. However, there are a few comments that I find important. I think it is a fair request made by Reviewer 1C4t to re-evaluate (or re-train) on VVT and ViViD. The authors indicate that their work targets "high-resolution, high-FPS Video Virtual Try-On", which requires higher data quality. While this may be true, the fact that the method is dependent so much on high-quality data is an important limitation. The authors only make some very high-level comment on this without providing any discussion or in-detail comment. Another comment made by reviewer wxd2 has only partially been addressed. The idea of using separate, simpler conditioners (“encoders”) is valid and it might be true what the authors write in their rebuttal but this should not have stopped then to actually try this and prove that indeed using a more complex architecture in necessary.

**Reviewer Scores:**

I think the discussion might have improved some of the opinions of the reviewers as indicated by Reviewer bChG and as claimed by the authors for Reviewer wxd2. Still, the comments of Reviewer 1C4t have only partially been addressed and this is critical given that this is the most negative reviewer. The main problem is that all the initial scores (with one exception) were rather negative and the concerns were real. This has been acknowledged also by the authors in their rebuttal. I read carefully the rebuttal and I think that the authors have done in several parts quite well. However, some of the initial concerns are still outstanding. Overall, I think the authors have done a good job in their rebuttal but the starting point was too low to change the final decision.

---

### Decision · Program_Chairs · 2026-01-26

Reject